# Can Large Language Models Integrate Spatial Data? Empirical Insights into Reasoning Strengths and Computational Weaknesses

**Bin Han,    Robert Wolfe,    Anat Caspi,    Bill Howe**
University of Washington, Seattle, Washington
{bh193,rwolfe3,billhowe}@uw.edu, {caspian}@cs.washington.com

## Abstract

We explore the application of large language models (LLMs) to empower domain experts in integrating large, heterogeneous, and noisy urban spatial datasets. Traditional rule-based integration methods are unable to cover all edge cases, requiring manual verification and repair. Machine learning approaches require collecting and labeling of large numbers of task-specific samples. In this study, we investigate the potential of LLMs for spatial data integration. Our analysis first considers how LLMs reason about environmental spatial relationships mediated by human experience, such as between roads and sidewalks. We show that while LLMs exhibit spatial reasoning capabilities, they struggle to connect the macro-scale environment with the relevant computational geometry tasks, often producing logically incoherent responses. But when provided relevant features, thereby reducing dependence on spatial reasoning, LLMs are able to generate high-performing results. We then adapt a review-and-refine method, which proves remarkably effective in correcting erroneous initial responses while preserving accurate responses. We discuss practical implications of employing LLMs for spatial data integration in real-world contexts and outline future research directions, including post-training, multi-modal integration methods, and support for diverse data formats. Our findings position LLMs as a promising and flexible alternative to traditional rule-based heuristics, advancing the capabilities of adaptive spatial data integration.

## 1 Introduction

In addition to their language generation capabilities, Large Language Models (LLMs) provide intuitive natural language interfaces for advanced computational tasks, including code generation (Chen et al., 2021), data manipulation (Hassani & Silva, 2023), and context understanding (Ethayarajh, 2019). These capabilities reduce dependence on software engineers for domain experts, democratizing computational solutions into under-resourced domains.

In this context, we are interested in supporting experts in the spatial domain to perform macro-scale spatial data *integration* — the process of integrating multiple datasets from different sources into a higher quality dataset (Mustière & Jolivet, 2021). We particularly target scenarios involving network data representing roads and pedestrian pathways (Sereshgi & Wenk, 2024), where individual sources, though sparse, biased, and erroneous, offer complementary information when integrated effectively (Cavazzi et al., 2023). A core primitive for integration is to match two elements from different datasets, which is traditionally achieved through either optimization and machine learning-based methods or heuristic approaches rooted geometrical properties (Sereshgi & Wenk, 2024). Rule-based heuristic approaches handle common, predictable spatial patterns (Chen & Walter, 2010; Kong & Yang, 2019) but struggle to address the variety of edge cases found in practice (Zhonglianga & Jianhuaa, 2008; Qiu et al.). Optimization and machine learning methods, while potentially more robust, require extensive labeled data and complex problem formulation.

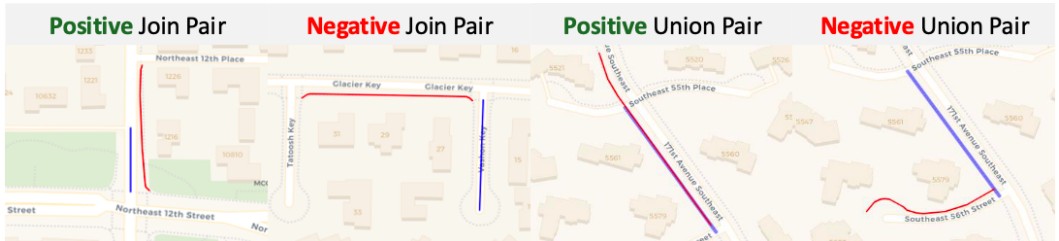

Figure 1: Positive and negative examples for the spatial join and union task. **(1)** For the spatial join task, the blue line represents the road, and the red line represents the sidewalk. The task classifies whether a sidewalk "runs alongside" a road from a pedestrian's perspective. **(2)** For the union task, both lines represent sidewalks. The task determines whether two sidewalks fully or partially represent the same real-world entity.

In this study, we explore the potential of LLMs to perform paired-elements matching based on natural language specifications provided by domain experts, and without the need for labeled data. Effective matching relies not only on computational geometry primitives (Jensen et al., 2004; Tristán et al., 2013) but also on a common sense understanding of the built environment. LLMs, pre-trained on vast diverse datasets (Dodge et al., 2021), possess the ability to discern semantic relationships and infer implicit patterns (Le Mens et al., 2023), making them potentially useful for reasoning about spatial data represented in text-based formats (e.g., GeoJSON). However, their ability to interpret natural language specifications as computational geometry problems and then solve them remains unclear.

We examine two element matching tasks — *spatial join*, which associates pairs of elements according to real-world relationships, and *spatial union*, which determines whether two objects fully or partially represent the same real-world entity. In both tasks, the input comprises candidate pairs of elements. Each element is a *linestring*, a sequence of contiguous edges connected end-to-end. The output is an indication of whether these pairs match according to the user-specified semantic relationship (e.g., a sidewalk adjacent to a road for join task) or whether they represent the same spatial feature (union task) (Figure 1).

We evaluate three heuristic-based baselines on both tasks. Heuristic methods identify relevant features derived from the pair (e.g., minimum angle between pairs of sub-edges, or minimum distance between points) and an appropriate threshold for each feature. We test against heuristic-based methods rather than broader machine learning-based methods for three reasons — First, heuristics are effective on both tasks, allowing us to compare LLM's ability. Second, simpler heuristic methods help isolate and clearly identify the fundamental limitations of LLMs in spatial reasoning tasks. Third, domain experts can easily express matching criteria using natural language rather than formulating complex algorithms.

We assess five LLMs using two methods — **(1) Heuristic prompting method:** We first assess LLM performance using zero-shot and few-shot natural language prompts, directly instructing models to generate labels without further explanation. We then incorporate heuristic guidance in the prompt, first in natural language (*hints*) and then as pre-computed numeric quantities (*features*). We find that natural language alone performs poorly even with hints (<60% accuracy), due to frequent computational and logical errors in identifying features or thresholds. But when pre-computed features are included, sidestepping the spatial reasoning task, the models are able to derive an appropriate threshold based on understanding, and significantly improves performance (>90% accuracy). This method is valuable in practice — reducing reliance on precise threshold specifications from prior data or domain knowledge, and enabling domain experts to run task-agnostic computations. **(2) Review-and-refine prompting method:** Motivated by self-correction research in LLMs, we employ a prompting approach where models first receive an initial answer (from heuristics, random guesses, or other methods) and are then prompted to review and refine it. This two-step method consistently improves poor initial answers and preserve accurate answers, enhancing overall accuracy. Overall, our contributions are as follows:

**1** We conduct comprehensive empirical evaluations of LLMs on spatial integration tasks, an area that, to our knowledge, remains largely underexplored. Our study serves as an initial step toward understanding how LLMs can be effectively used in spatial applications.

**2** We demonstrate that, while LLMs struggle given natural language instruction alone, prompting with relevant geometrical features derived from heuristics significantly enhances performance, reaching peak accuracies of 98.4% and 96.0% for each task, respectively.

**3** We perform qualitative analysis of spatial reasoning on real-world geometries instead of abstract shapes and textbook math problems, better reflecting practical challenges and new opportunities. The analysis reveals that while LLMs possess spatial reasoning abilities, they tend to commit logical and computational errors.

**4** We propose a two-step review-and-refine prompting strategy that consistently enhances poor initial answers (average improvements of 38% and 7.4% on the two tasks) while preserving the accuracy of strong initial answers. This approach reaches best accuracies of up to 99.5% and 96.5%, surpassing the performance of the most effective heuristic methods.

**5** We discuss the advantages and limitations of employing LLMs for spatial data integration tasks. Furthermore, we outline future research to address these limitations and enhance the applicability of LLMs for spatial data applications.

## 2 Related Work

**Generative Language Models & Diverse Applications:** We study autoregressive LLMs pretrained on large text datasets, particularly those instruction fine-tuned for natural, chat-based interactions (Wei et al., 2021; Ouyang et al., 2022). LLMs have been studied and applied across diverse domains, including medicine (Ramachandran et al., 2023; Han et al., 2023), finance (Wu et al., 2023; Kong et al., 2024), law (Jiang et al., 2024; Louis et al., 2024), education (Zhang et al., 2024; Dangol et al., 2024), software development (Zheng et al., 2024; Lin et al., 2024), behavioral analysis (Chiu et al., 2024; Xu et al., 2025), translation (Koshkin et al., 2024; Han et al., 2025), and fact-checking (Das et al., 2023; Wolfe & Mitra, 2024). Our research focuses on urban analytics, building upon prior work by (Wang et al., 2024), who developed personalized urban mobility strategies using LLMs, and (Han et al., 2024), who introduced a zero-shot urban annotation framework with vision-language models (VLM).

**Self-Correction Using LLMs:** Recent studies have demonstrated that prompting LLMs to self-correct can substantially enhance task performance (Madaan et al., 2023; Kim et al., 2023; Kamoi et al., 2024). The core idea involves an LLM initially generating an output; and subsequently using the same model to iteratively provide feedback and refine the initial output. Similarly, cross-model correction methods utilize feedback provided by different models than those that generated the initial responses (Cohen et al., 2023; Li et al., 2023; Liang et al., 2023). Importantly, both correction strategies eliminate the need for supervised training or fine-tuning. A related concept, the **"LLM-as-a-Judge"** paradigm, uses LLMs to evaluate or rank outputs across various applications, including retrieval (Li & Qiu, 2023), alignment (Lee et al., 2023), and reasoning (Zhao et al., 2024). Detailed surveys are available for further exploration (Kamoi et al., 2024; Li et al., 2024a;b).

**Spatial Data Integration:** *Spatial data join* merges two spatial datasets by evaluating geographic features. Rule-based methods include geometrical and distance-based joins (Jacox & Samet, 2007). Geometrical joins utilize qualitative spatial predicates (e.g., intersects, contains, overlaps) to link features (Hope & Kealy, 2008), while distance-based joins rely on quantitative distance measurements. *Spatial data union*, a spatial overlay operation, combines multiple spatial datasets into one comprehensive dataset covering the combined area (Margalit & Knott, 1989) Our study differs from prior research in that — rather than optimizing efficiency or scalability for large-scale datasets, using known rules (Du et al., 2017; Vu et al., 2024), we address ambiguous scenarios requiring human judgment. Additionally, we use LLMs to evaluate joinability and unionability without explicit rules.

## 3 Tasks & Datasets

Spatial join and union tasks integrate datasets from heterogeneous geospatial resources. In our context, the two tasks determine whether two geometric objects should be linked based on specific criteria. Denote $\mathcal{S}_i = \{s_1, s_2, \ldots, s_n\}$ as a LineString geometry representing a

sidewalk annotation, connecting each pair of points $(s_i, s_{i+1})$ with a line. We will refer to each sub-line as an *edge*. Similarly, let $\mathcal{R}_i = \{r_1, r_2, \ldots, r_m\}$ to represent a road annotation.

The **Spatial Join** task classifies whether a sidewalk "runs alongside" a road from a pedestrian's perspective. This relationship is not precisely defined, but is important for screen readers used by vision impaired travelers. If the reader makes a mistake, the audible directions could be misleading or in the worst case dangerous: the road should be called out when sighted travelers would tend to agree that they are walking along the road. Computationally, this relationship corresponds, approximately, to adjacency and parallelism between road and sidewalk, defined by a binary classification function $f_\theta(\mathcal{S}_i, \mathcal{R}_i) \to \{0, 1\}$. The **Spatial Union** task determines whether two sidewalk annotations from different sources, $\mathcal{S}_i$ and $\mathcal{S}'_i$, refer fully or partially to the same sidewalk, defined similarly as $f_\theta(\mathcal{S}_i, \mathcal{S}'_i) \to \{0, 1\}$. Complications include partial matches and errors in one or both sources, and heterogeneity in different built environments and relevant laws.

We chose these two tasks because: **(1)** while they appear simple, they are foundational to more complex spatial analysis, such that focusing on these ubiquitous, fundamental tasks provides us with a highly interpretable view of LLMs' basic capabilities in handling spatial tasks, and helps us identify the strengths and weaknesses of LLMs in working with spatial data; and **(2)** a small number of heuristic rules perform well for these two tasks, which allows us to test LLM spatial reasoning in a controlled way. In cases where heuristics are insufficient and do not generalize well, we can assess LLMs ability on those specific cases.

We obtained two datasets from the Transportation Data Exchange Initiative (Initiative, 2025): **(1)** Bellevue-Sidewalk, comprising 14,200 sidewalk-specific annotations of attributes such as footway type and geometry; and **(2)** Bellevue-City, containing 73,700 annotations of both roads and sidewalks sourced from OpenStreetMaps, annotated with footway type, highway type, and geometry. To construct prediction datasets for our analysis, we implemented the pre-processing steps outlined in Appendix A, allowing us to split the data into train, validation, and test sets for evaluation. Ultimately, for the join task, we have 6,442; 716; and 1,000 $(\mathcal{S}_i, \mathcal{R}_i)$ training, validation and test pairs. For the spatial union task, we have 837, 93, and 399 $(\mathcal{S}_i, \mathcal{S}'_i)$ pairs. Figure 1 illustrates a positive and a negative example for each task: the candidate pair are near each other, but do not satisfy the semantic conditions. For all object pairs in both datasets, we compute three features based on geometrical relationships between the objects (which are predictably and demonstrably relevant to the task):

- `min_angle`: the smallest angle (in degrees) between two spatial objects, computed by dividing each annotation into smaller segments, calculating pairwise acute angles between segments, and selecting the smallest angle as an indication of object alignment.
- `min_distance`: the smallest distance (in meters) between any point on one annotation and any point on the other annotation. This metric reflects the proximity between geometries.
- `max_area`: the percentage of overlapping area between two objects, calculated after applying a 10-meter (see Appendix A for selection rationale) buffer around each object. This metric reflects the potential similarity between geometries.

Appendix B describes the distributions of all geometrical features in both datasets.

## 4   Methods

In this section, we describe the heuristic baselines (§4.1), our proposed heuristic prompting method (§4.2), and the review-and-refine method (§4.3). In addition, we describe the experimental setup and language models assessed in our study in §4.4.

### 4.1   Baseline Heuristics

We use relevant heuristics commonly applied in the join or union task that leverage geometrical relationships between the pairs. Their effectiveness help us compare LLMs' ability.

**Parallelism (p):** evaluates whether two linestrings are approximately parallel, quantified by the minimum angle (`min_angle`) between the pairs. We test five thresholds for each task, $\alpha$

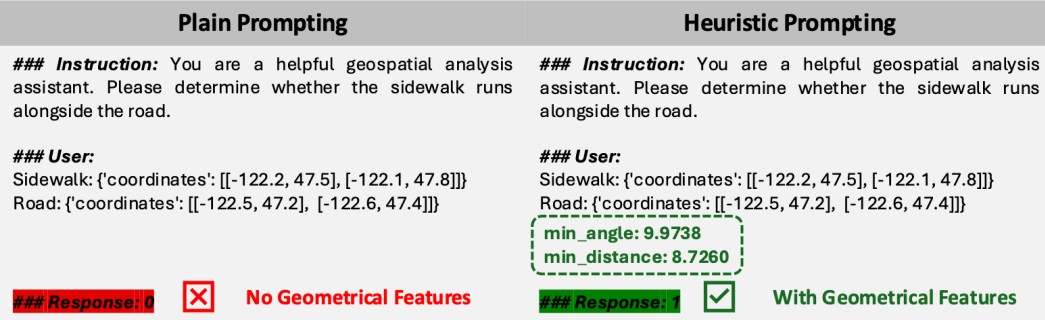

Figure 2: Plain prompting (left) and our heuristic prompting (right) for the spatial integration tasks. In the plain prompting, only the basic task description is provided. In the heuristic prompting, features derived from heuristics are included to further enhance performance.

= {1,2,5,10,20} degrees for the join task and $\alpha$ = {1,2,3,4,5} degrees for the union task. The heuristic is computed as: $\hat{y} = \mathbb{1}[\texttt{min\_angle} \leq \alpha]$, where $\alpha$ is a pre-defined threshold value.

**Clearance Heuristic (c):** assesses whether two linestrings maintain a minimum distance (`min_distance`) between them. For example, a sidewalk likely runs alongside a road if the distance between is above a given threshold. We test five thresholds $\{1, 2, 3, 4, 5\}$ exclusively in the join task (candidate pairs for union intersect, such that `min_distance` is always zero). The heuristic is computed as: $\hat{y} = \mathbb{1}[\texttt{min\_distance} \geq \alpha]$, where $\alpha$ is a threshold value.

**Overlap Heuristic (o):** determines the extent of overlap between linestrings using `max_area`. Two annotations likely represent the same object if they exhibit a high overlap ratio. We report $\{0.1, 0.2, 0.3, 0.4, 0.5\}$ percent thresholds for join and $\{0.5, 0.6, 0.7, 0.8, 0.9\}$ for union. The heuristic is computed as: $\hat{y} = \mathbb{1}[\texttt{max\_area} \geq \alpha]$

**Duo & Trio Heuristics:** We combine two or three heuristics for each task. We use abbreviations {(p), (c), (o)} to denote the parallel, clearance, and overlap heuristics respectively. For join, we evaluate three duo heuristics {(p,c), (p,o), (c,o)} and one trio heuristic {(p,c,o)}. For union, we evaluate one duo heuristic {(p,o)}. We use the same thresholds above.

## 4.2 Heuristic Prompting Method

As a baseline, we evaluate pre-trained LLMs using zero-shot and few-shot prompting, providing only a task description and the GeoJSON and requesting that the models generate binary labels without further explanation. In the few-shot scenario, we include one positive example and one negative example. We denote this baseline approach (zero- and few-shot) as (plain). We then extend the baseline by incorporating heuristic guidance to assist the LLM in solving the integration tasks. We consider two forms of heuristic guidance:

**Natural Language Hints (hints)**: Descriptions of one or a few heuristics in natural language.

**Geometrical Features (features)**: Descriptions of one or a few heuristics in natural language, and relevant geometrical features (e.g., `min_angle`, `min_distance`, `max_area`).

See Figure 2 for examples. We describe specific prompt formulations in Appendix §F.

## 4.3 Review-And-Refine Method

Heuristic performance varies significantly with threshold selection. We consider whether LLMs can automatically select an appropriate threshold without labeled data. That is, can we improve on a "bad" guess of a threshold using the parameterized knowledge of an LLM?

For each task, we generate initial responses using the best- and worst-performing heuristics, as well as random guesses. Following Kim et al (Kim et al., 2023), we provide initial guesses in a prompt and ask the model first to review the guess and identify potential issues. Then in a second pass, we include the review with the initial prompt and request a refined answer. Unlike Kim et al, we perform these steps only once rather than iteratively. All experiments are conducted under few-shot prompting, utilizing either heuristic hints or features.

### 4.4 Models & Experimental Setup

For our primary experiments, we evaluate five pretrained LLMs. These include two smaller, open-source models, {Meta-Llama-3.1-8B-Instruct (llama3), Mistral-7B-Instruct-v0.3 (mistral)}, and three larger proprietary, API-based models, {GPT-4o-mini, 2024-07-18 snapshot (4o-mini), Qwen-plus (qwen-plus), GPT-4o, 2024-08-06 snapshot (4o)}. Models were selected based on their scale, performance on public LLM leaderboards, and inference costs. For a qualitative ablation study, we also assess a proprietary, API-based reasoning model: Deepseek-R1. Reasoning models are post-trained using reinforcement learning for complex reasoning tasks, and they exhibit the capacity to improve their answers by drawing on internal chains of thought before responding to a user (Guo et al., 2025).

The experimental setups remain consistent across models and two tasks. For llama3 and mistral, we employ Unsloth (Daniel Han & team, 2023) for accelerated inference with 4-bit quantization, configuring *max_sequence_length* and *top_p* to 4096 and 10, respectively. For proprietary models, *temperature* and *top_p* are set to 0 and 1. In the heuristic-driven prompting method, all models generate 10 new tokens per prompt. For the review-and-refine method, each model produces 500 new tokens per example. Outputs are post-processed and evaluated using accuracy as the primary metric.

## 5 Results

In this section, we present detailed results for both the spatial join and union tasks. We structure these findings into *takeaways* to facilitate understanding, denoted as **T#**

### 5.1 (T1) Baseline heuristics can be highly effective, but performance depends on task-specific selection and meticulous tuning of feature thresholds.

Summary of baseline heuristic results for both tasks are presented in Table 1. More detailed heuristic results are presented in Appendix C. We observe that these tasks are amenable to effective heuristic solutions: For the join task, the best single heuristic achieves 94.8% accuracy, whereas duo and trio heuristics achieve 98.7% and 99.0%. We observe a similar trend for the union task, with duo heuristics outperforming single heuristics.

Despite their effectiveness, the ability for target users to select good heuristics and good thresholds cannot be assumed, as they vary from task to task. For example, the worst-performing join feature (max_area overlap) is up to 24% lower accuracy than the best (min_angle parallelism), even with good thresholds. For the best feature, the difference in accuracy between the best and worst threshold is 33.6%. Similar observation applies to the union task. Finally, the best choice of feature and threshold vary from task-to-task suggesting pressure on users to develop intuition about every possible integration scenario. Even with user expertise, practically identifying the optimal configuration remains a challenging task likely to involve trial and error.

| heuristics | spatial join | | | spatial union | | |
|---|---|---|---|---|---|---|
| | worst | best | avg (std.) | worst | best | avg (std.) |
| single | 0.654 | 0.948 | 0.822 (0.10) | 0.842 | 0.927 | 0.880 (0.03) |
| duo | 0.735 | 0.987 | 0.910 (0.06) | 0.870 | 0.962 | 0.933 (0.02) |
| trio | 0.715 | 0.990 | 0.904 (0.08) | - | - | - |
| overall | 0.654 | 0.990 | 0.900 (0.08) | 0.842 | 0.962 | 0.918 (0.04) |

Table 1: Baseline heuristic results for the spatial join and union tasks. The worst, the best, and the average accuracy (standard deviation) are reported for single, duo, and trio (spatial join only) heuristics. Standard deviations are calculated given different thresholds. For the spatial join task, duo results are averaged across all three duo combinations.

### 5.2 (T2) While spatial integration tasks inherently challenge LLMs, incorporating heuristic features significantly enhances performances.

Average results of heuristic-driven prompting for join and union are visualized in Figure 3. Detailed numerical results are provided in Tables 9 and 10 in Appendix D, respectively.

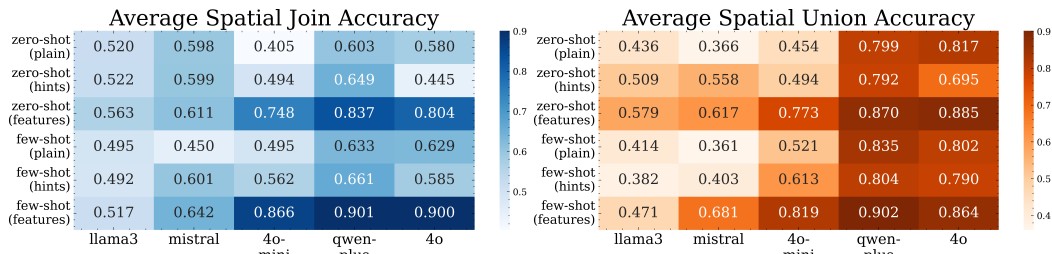

Figure 3: Average results of heuristic prompting method for the spatial join and union task. Each row represents prompting with no heuristic (plain) or different heuristic information (hints or features), under either zero- or few-shot prompting.

**Prompting with natural language (plain), even with feature hints (hints), is not effective.** The models achieve results comparable to the worst heuristics. For the join task, the five models reach average accuracies of 54.2% and 57.5% under zero- and few-shot prompting, respectively, which are 39.8% and 36.1% lower than the overall heuristic average accuracy. Likewise, for union, the models attain average accuracies of 60.1% and 59.6%, 34.6% and 35.1% lower than heuristic averages. These results indicate that models struggle to understand the scenario, translate the scenario into a computational problem, or accurately solve the problem. We investigate reasoning in Section 5.3.

**Prompting with heuristic features is remarkably effective.** The availability of pre-computed features may afford sidestepping the computational geometry and provide a more direct link between the natural language specification and the identification of an appropriate threshold. We observe that — **1** *Larger proprietary models deliver the most competitive performance.* For the join task, smaller open models (llama3 and mistral) exhibit low accuracy, suggesting their limitations for spatial reasoning. The average accuracies of qwen-plus (90%) and 4o (90.1%) match the overall heuristic baseline (90%), with peak performance of 98.4% (qwen-plus) and 97.4% (4o) from Table 9. Qwen-plus aligns closely with the best-performing duo heuristic (98.7%) and nearly matches the top trio heuristic (99.0%). Similar trends appear in the spatial union task, with qwen-plus and 4o achieving peak accuracies of 96.0% and 95.5%, respectively, closely matching the best duo heuristic (96.2%). The subsequent observations focus exclusively on these proprietary models. **2** *Prompting with more heuristic features improves accuracy.* Table 9 shows that min_distance or max_area heuristics are ineffective on their own but combinations improve performance. While this result is unsurprising, it provides evidence of LLMs deriving rule-based systems. **3** *LLMs eliminate the need for feature selection and threshold tuning.* The selection of different features and corresponding thresholds significantly mediates heuristic-based performance; LLMs are effective in selecting both feature and threshold based on their spatial awareness. For the spatial join task, there are 215 possible heuristic combinations (15 single, 75 duo, and 125 trio combinations). The highest qwen-plus accuracy (98.4%) surpasses 93% of these heuristic combinations, while the top 4o model accuracy (97.4%) outperforms 85%.

## 5.3 (T3) LLMs exhibit fundamental spatial reasoning abilities, but struggle to derive and solve the relevant computational geometry.

We consider how LLMs approach the element matching task in order to understand why providing numeric features is required for competitive performance. For the join task, we employ few-shot, chain-of-thought (CoT) prompting on the {4o-mini, qwen-plus, 4o} models, using the standard "step by step" CoT instruction and setting the maximum generation tokens to 2,000. We inspected the outputs across 20 samples, under no heuristics (plain), with hints, and with features. We observe that:

**1** *Models exhibit spatial reasoning capabilities.* Even without heuristic features, models understand the task by evaluating spatial conditions such as proximity (distance) and alignment (angle or orientation) between sidewalk-road pairs. Specifically, model 4o-mini mentions {*proximity*, *distance*} in 19 of 20 instances and {*alignment*, *angle*} in 5 of 20 instances (20/20 and 19/20, respectively, for qwen-plus; 20/20 and 11/20 for 4o). These findings

suggest LLMs can identify spatial concepts relevant to the task. However, **2** *models are ineffective at computational geometry.* All three models interpret the spatial join primarily as a computational task involving geometrical calculations but consistently fail to execute these correctly. Errors fall into three categories: (1) *Incorrect computational logic*, with fundamentally flawed reasoning; (2) *Unclear computational explanations*, characterized by incomplete or vague details, often indicated by terms like {*appears, approximately*}; and (3) *Conclusions without computation*, where results are asserted without computational backing, signaled by keywords like {*assume, appear*}. Examples of these errors are shown in Table 11 in Appendix E. Occasionally, correct conclusions occur by chance despite erroneous computations. **3** *Providing heuristic features fundamentally changes the approach to the task.* Including heuristic features transforms the spatial join task from a computational geometry problem into a simpler heuristic evaluation problem, where models verify heuristic conditions based on provided values. It significantly reduces task difficulty and enhances model performance. We also notice that — First, models typically apply reasonable heuristic thresholds (*e.g.*, 10 degrees for angles, 5 meters for distance, 30% for overlap). However, some thresholds do not clearly differentiate between provided positive and negative examples, indicating that decisions are not solely data-driven, but more relying on their inherent understanding of the task and real-world context. Second, thresholds chosen by models vary considerably across examples, ranging from 2–10 meters for distance and 20%–70% for overlap. This variability indicates that LLMs employ a flexible heuristic approach with potentially distinct thresholds per sample rather than a fixed universal threshold.

We conducted the same CoT prompting on 20 samples using Deepseek-R1, a reasoning model known for generating detailed reasoning steps. We avoid processing all 1000 samples to reduce costs, computational time, manual review time, and because the patterns began to converge. We observed that — **1** *Deepseek-R1 exhibits strong computational geometry capabilities.* Similar to other models, Deepseek-R1 interprets the task computationally without heuristic features. However, Deepseek-R1 performs precise geometrical calculations, including point-by-point and segment-by-segment evaluations, occasionally utilizing advanced measures like the Haversine distance. **2** *However, its conclusion logic is fundamentally flawed.* Despite advanced computational skills, Deepseek-R1 employs unreliable logic when arriving at conclusions. For instance, to determine parallelism, it cites evidence such as "The road's vector is southeast, and sidewalk's vector is northeast; thus, they are not parallel." To assess proximity, it references questionable logic such as "The sidewalk's start is 23 meters west of the road's start, beyond typical adjacency for sidewalks." This suggests that even models with powerful computational skills may struggle with spatial reasoning and logical coherence essential for spatial integration tasks.

### 5.4 (T4) Review-and-refine method is highly effective — significantly improving poor initial answers while preserving good ones.

Figure 4 presents the results of the review-and-refine method for both spatial integration tasks. We observe that — **1** *Review-and-refine with only hints is ineffective.* As previously discussed in §5.2, direct prompting with natural language hints, yields poor results. This trend continues here — in the spatial join task, average accuracy for 4o-mini, qwen-plus, and 4o decreases by 47.9%, 17.8%, and 8.1%, respectively, from initial responses, indicating that heuristic hints alone are insufficient for effective revisions. **2** *Review-and-refine with heuristic features is highly effective.* Providing heuristic features significantly improves poor initial responses and maintains or enhances strong answers. For the join task, poor initial responses (random, worst heuristics), the three models achieve average accuracy improvements of 36.4%, 35.9%, and 41.8%, respectively. When initial answers are strong, qwen-plus and 4o maintain or increase accuracy. Notably, 4o surpasses the best heuristic baseline accuracy (99%) by an additional 0.4%. For the spatial union task, when initial answers are poor, the three models improve average accuracy by 5.2%, 9.2%, and 7.7%, respectively.

**How does the review-and-refine process make decisions?** **1** Without features, the review-and-refine process aims to solve the task using computational geometry. With heuristic features included, the review-and-refine process systematically checks heuristic conditions against provided features, significantly reducing task difficulty and improving performance.

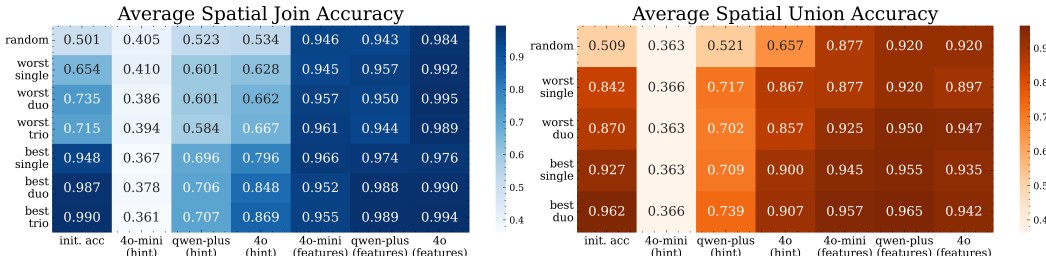

Figure 4: Accuracies of review-and-refine method on both spatial join and union task. Each row represents different ways of generating initial answers, either randomly or using heuristics. We report accuracies when heuristic hints or features are provided.

Results demonstrate that LLMs perform well regardless of the initial answer quality, even outperforming the best heuristic. **2** *Initial answers variably influence final results.* Review-and-refine results remain relatively stable and consistent for 4o-mini (95.5%) and 4o (98.9%), indicating these models remain unaffected by initial answers. However, refined results from poor initial answers for qwen-plus (average 94.9%) are notably lower compared to results derived from strong initial answers (average 98.4%). A manual review of several correction examples reveals that qwen-plus typically considers overlap thresholds (max_area) between 0.1 and 0.6 sufficient for positive labels. However, with an initial negative answer (0), the review process sometimes judges a threshold around 0.25 as insufficient.

# 6 Ablation Studies

In this section, we present two ablation studies. In Section 6.1, we study spatial integration using vision-language models (VLMs), where visual information is incorporated as contextual input. In Section 6.2, we provide results from applying LLMs to four additional spatial analysis tasks, providing insights on LLMs' generalization in the spatial context.

## 6.1 Spatial Integration With Vision-Language Models

In our core experiments, we relied solely on textual information extracted from GeoJSON files. However, visual information may also assist models in spatial integration tasks, as images provide direct cues about orientation and proximity. To explore this possibility, we conducted preliminary experiments using vision-language models (VLMs) on the two integration tasks. We evaluated one open-source model, {Qwen-2.5-VL-7B-Instruct (qwen-vl)}, and two API-based models, {gpt-4o-mini and gpt-4o} in a zero-shot setting, supplying only the plain task description with an image input. The results are summarized in Table 2. We observe two key findings: (1) Incorporating visual information improves performance on spatial integration tasks compared to text-only prompting, demonstrating the effectiveness of visual cues and (2) heuristic prompting using geometric features (features) still outperforms prompting with visual inputs. These results suggest that while visual context is beneficial, explicitly providing geometric features remains more effective for these tasks.

| | spatial join | | | spatial union | | |
|---|---|---|---|---|---|---|
| models | plain | plain+img | features | plain | plain+img | features |
| qwen-vl | - | 0.566 | - | - | 0.383 | - |
| 4o-mini | 0.405 | 0.627 | 0.938 | 0.454 | 0.624 | 0.797 |
| 4o | 0.580 | 0.728 | 0.948 | 0.817 | 0.889 | 0.955 |

Table 2: Plain prompting (plain) only uses textual information, while plain+img prompting presents an additional satellite image as visual context. Heuristic prompting with geometric features still has the best performance, indicating the importance of geometrical information.

## 6.2 Diverse Spatial Reasoning Tasks

Beyond evaluating LLMs on the two core spatial integration tasks, we extend our analysis to a broader set of spatial reasoning tasks on real geometry objects. These additional experi-

ments aim to provide more general insights into the spatial reasoning and computational capabilities of LLMs. We focus on four tasks that differ in geometry types, computational complexity, and reasoning logic, thereby broadening the evaluation scope while remaining grounded within the spatial analysis domain:

- Point-to-Seg. Dist (P2S): Compute the shortest distance between a point and a segment.
- Spatial Containment (SC): Determine whether a point lies inside a polygon (triangle).
- Spatial Intersection (SI): Check if two line segments intersect.
- Convex Hull (CH): select the minimal set of points forming the convex hull of a point set.

Each task was evaluated on 50 synthetic geometries samples defined within a unit square. We tested the same API-based models {qwen-plus, 4o-mini, and 4o}, and report accuracy. For P2S, a tolerance of 1e-3 was used for correctness. Results are shown in Table 3.

We observe that the models perform well on simpler tasks (P2S: 90%, SC: 81%) but show notable performance drops on more complex tasks (SI: 70%, CH: 57%). A brief qualitative analysis of GPT-4o's outputs revealed promising indications of spatial reasoning: the model consistently applied projection logic for P2S, ray casting for SC, orientation-based logic for SI, and referenced algorithms such as Graham's scan or Jarvis March for CH. However, inaccurate computations still frequently led to incorrect results. These findings are consistent with our earlier observations from the integration tasks. Collectively, the results suggest that while LLMs demonstrate spatial reasoning capabilities that extend beyond basic integration, they still face challenges in executing the precise computations required for general spatial analysis tasks.

| models | P2S | SC | SI | CH |
|---|---|---|---|---|
| qwen-plus | 0.94 | 0.82 | 0.80 | 0.62 |
| 4o-mini | 0.82 | 0.76 | 0.64 | 0.46 |
| 4o | 0.94 | 0.84 | 0.66 | 0.64 |

Table 3: LLMs' performances on four additional spatial tasks. Accuracies are reported.

## 7 Conclusions

We have observed effective and robust performance from LLMs on spatial integration tasks. We find that LLMs are generally unreliable at translating task prompts in natural language into solvable computational geometry problems. When provided with numeric features, LLMs can infer suitable thresholds on a case-by-case basis and achieve competitive results. Chain-of-thought analyses reveal that models often sidestep explicit reasoning by relying on world knowledge to estimate thresholds. The review-and-refine method improves suboptimal outputs while preserving accurate ones, sometimes even surpassing the best heuristics. Overall, spatial data integration is a promising application area for LLMs, though current models remain inadequate for tasks requiring rigorous computational geometry-based reasoning. We outline the following potential future directions — **(1) Post-Training.** Implementing a post-training regime to enhance model comprehension of the built environment and computational geometry primitives, advancing natural language interfaces for complex conflation tasks. **(2) Multi-Modal Approaches.** We have demonstrated that integrating visual modalities through vision-language models could improve performance. More advanced methods of using the visual information could be explored. **(3) Task-Agnostic Approaches.** Expanding beyond GeoJSON and LineString geometries to develop unified methodologies supporting diverse spatial data formats (e.g., Shapefile, GeoTIFF, KML) and geometry types (Points, Polygons) would greatly enhance the versatility and applicability of LLM-based geospatial approaches.

## 8 Acknowledgments

This work was supported by the U.S. Department of Transportation (USDOT) Intelligent Transportation Systems for Underserved Communities (ITS4US) Deployment Program, administered by the Joint Program Office (JPO), under Cooperative Agreement Number 693JJ32250014. Anat Caspi was additionally supported by the Taskar Endowment Fund from the Taskar Center for Accessible Technology.

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

# A  Data Processing

**Spatial Join Dataset** We first filter roads from Bellevue-City, selecting only those with highway types in {secondary, residential, tertiary, primary, living_street} (types likely to have paired sidewalks), resulting in 17,800 road annotations. We add a 10-meter buffer around each road. The 10-meter buffer reflects the origins of our dataset in accessibility research, which is typically concerned only with roads that have sidewalks. Thus, while there is substantial variation in real-world road widths, our dataset includes only road types likely to have adjacent sidewalks – primary, secondary, tertiary, residential, living_street, and excludes highways. The selection of the 10-meter buffer is based on typical road widths in Seattle[1] and Bellevue[2], and applied across all types.

Next, we perform a spatial join operation between the buffered roads and sidewalks using an intersection predicate. This step generates 14,556 potential $(\mathcal{S}_i, \mathcal{R}_i)$ candidate pairs, under the assumption that a sidewalk running alongside a road should fall within the buffer zone. Each $(\mathcal{S}_i, \mathcal{R}_i)$ pair was then manually labeled through visual inspection, resulting in 6,034 positive pairs and 4,206 negative pairs. We split the labeled dataset into training, validation, and test sets, containing 6,442; 716; and 1,000 samples, respectively. The remaining pairs were deemed ambiguous and required case-by-case discussion; they were thus excluded from this study.

**Spatial Union Dataset** We use the 17,100 sidewalk annotations from Bellevue-City. A spatial join operation is performed between the two sources of sidewalk annotations using an intersection predicate. This step generates 5,496 $(\mathcal{S}_i, \mathcal{S}'_i)$ unionable candidate pairs, under the assumption that if two annotations represent the same sidewalk, either fully or partially, they should overlap. Each pair was manually labeled through visual inspection, resulting in 886 positive pairs and 443 negative pairs. The final dataset was split into training, validation, and test sets, containing 837; 93; and 399 samples.

**Disclaimer: (1)** Our spatial join and union task represent the generic spatial integration tasks. There could be hundreds of specific tasks given users' needs. In our study, we use the two generic ones for research purpose — to demonstrate the feasibility and effectiveness of using LLMs in spatial integration application. **(2)** The candidate pairs generation process uses certain spatial conditions (*e.g.*, 10-meter buffer for join data and intersection for union data). We use those loose conditions to narrow down the search space for the potential pairs. We are not checking all possible pairs.

---

[1] https://streetsillustrated.seattle.gov/street-type-standards/downtown/.

[2] https://bellevuewa.gov/sites/default/files/media/pdf_document/trans-design-manual-design-standards-2017.pdf.

# B   Geometrical Feature Distributions

In this section, we present the distributions of the geometric features for each task. For the spatial join task, we plot `min_angle`, `min_distance`, and `max_area`. For the spatial union task, we plot `min_angle` and `max_area`. The distributions are visualized based on the training sets. We also show the accuracies of applying different threshold values on the training data, which helps us select appropriate thresholds to apply to the test sets.

## B.1   Spatial Join Task

**Parallel Heuristic (`min_angle`):** Figure 5 shows the distribution of `min_angle` in the training set from the spatial join dataset. We observe that: (1) most positive pairs have small angle degrees. While most negative pairs have larger degrees, some have small degrees, making them false positive predictions if the threshold value is large. 2) `min_angle`=7 gives the best train accuracy (93.76%), indicating that it is a good heuristic/feature for the spatial join task.

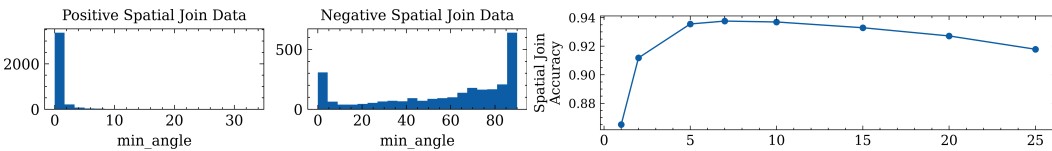

Figure 5: **Left:** Distribution of `min_angle` in positive and negative spatial join data. **Right:** train accuracy using different feature thresholds for `min_angle`, for spatial join task.

**Clearance Heuristic: (`min_distance`)** Figure 6 shows the distribution of `min_distance` in the training set from the spatial join dataset. We observe that: (1) most negative pairs have smaller values because they tend to overlap or intersect with each other. (2) `min_distance`=3 gives the best train accuracy (86.57%), indicating that it is an ordinary heuristic for the spatial join task.

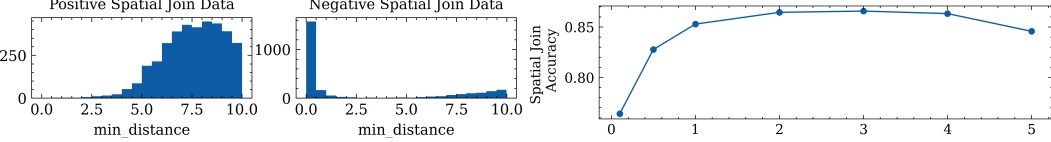

Figure 6: **Left:** Distribution of `min_distance` in positive and negative spatial join data. **Right:** train accuracy using different feature thresholds for `min_distance`, for spatial join task.

**Overlap Heuristic (`max_area`):** Figure 7 shows the distribution of `max_area` in the training set from the spatial join dataset. We observe that: (1) both positive and negative pairs have similar distribution when `max_area` is greater than 0.2. But negative pairs have way more smaller overlap values. (2) `max_area`=0.4 gives the best train accuracy (71.00%), indicating that it is not a good heuristic for the spatial join task.

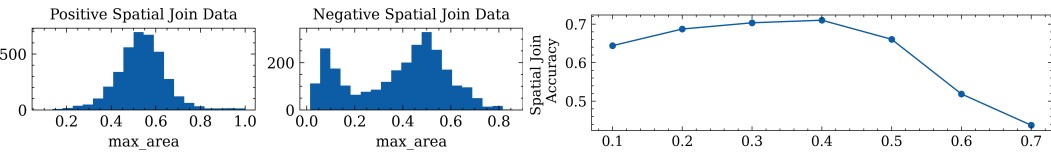

Figure 7: **Left:** Distribution of `max_area` in positive and negative spatial join data. **Right:** train accuracy using different feature thresholds for `max_area`, for spatial join task.

### B.2 Spatial Union

**Parallel Heuristic (`min_angle`):** Figure 8 shows the distribution of `min_angle` in the training set from the spatial union dataset. We observe that: (1) positive and negative pairs have similar distributions, while negative pairs have a few large degrees. 2) `min_angle=2` gives the best train accuracy (85.66%), indicating that it is an ordinary heuristic for the union task.

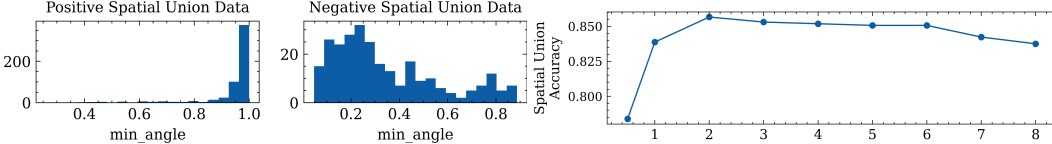

Figure 8: **Left:** Distribution of `min_angle` in positive and negative spatial union data. **Right:** train accuracy using different feature thresholds for `min_angle`, for spatial union task.

**Overlap Heuristic (`max_area`):** Figure 9 shows the distribution of `max_area` in the training set from the spatial union dataset. We observe that: (1) most positive pairs have large overlap percentage values. While most negative pairs have small overlap percentages, some still have large values. (2) `max_area=0.8` gives the best train accuracy (92.83%), indicating that it is a good heuristic for the union task.

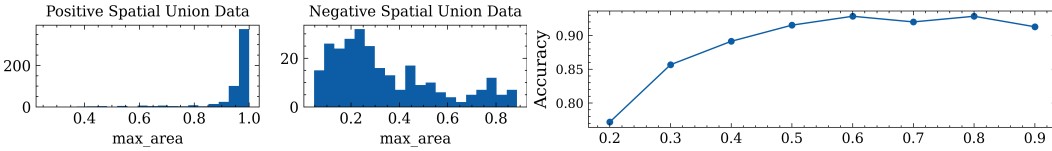

Figure 9: **Left:** Distribution of `max_area` in positive and negative spatial union data. **Right:** train accuracy using different feature thresholds for `max_area`, for spatial union task.

# C  Baseline Heuristic Results

In this section, we present the detailed results of heuristic baselines for each task. For all tables under each task, the results are color-coded with the same min-max values, with darker shades corresponding to higher accuracy. The best configurations of heuristics are described in each table's caption, with the best results colored blue.

## C.1  Spatial Join Heuristic Results

| parallel (min_angle) | Accuracy (acc) | clearance (min_distance) | Accuracy (acc) | overlap (max_area) | Accuracy (acc) |
|---|---|---|---|---|---|
| 1 | 0.867 | 1 | 0.838 | 0.1 | 0.655 |
| 2 | 0.908 | 2 | 0.847 | 0.2 | 0.701 |
| 5 | 0.946 | 3 | 0.851 | 0.3 | 0.735 |
| 10 | 0.948 | 4 | 0.852 | 0.4 | 0.753 |
| 20 | 0.934 | 5 | 0.835 | 0.5 | 0.654 |

Table 4: Single heuristic results for the spatial join task. The best single heuristic configuration is parallel (min_angle=10).

| parallel (min_angle) | clearance (min_distance) 1 | 2 | 3 | 4 | 5 |
|---|---|---|---|---|---|
| 1 | 0.868 | 0.868 | 0.866 | 0.865 | 0.847 |
| 2 | 0.910 | 0.910 | 0.908 | 0.907 | 0.885 |
| 5 | 0.948 | 0.948 | 0.946 | 0.945 | 0.922 |
| 10 | 0.951 | 0.951 | 0.949 | 0.948 | 0.925 |
| 20 | 0.941 | 0.941 | 0.939 | 0.938 | 0.915 |

| parallel (min_angle) | overlap (max_area) 0.1 | 0.2 | 0.3 | 0.4 | 0.5 |
|---|---|---|---|---|---|
| 1 | 0.883 | 0.889 | 0.888 | 0.864 | 0.735 |
| 2 | 0.930 | 0.938 | 0.937 | 0.906 | 0.765 |
| 5 | 0.970 | 0.981 | 0.979 | 0.944 | 0.781 |
| 10 | 0.974 | 0.987 | 0.987 | 0.952 | 0.782 |
| 20 | 0.962 | 0.981 | 0.986 | 0.950 | 0.781 |

| clearance (min_distance) | overlap (max_area) 0.1 | 0.2 | 0.3 | 0.4 | 0.5 |
|---|---|---|---|---|---|
| 1 | 0.890 | 0.930 | 0.958 | 0.938 | 0.775 |
| 2 | 0.906 | 0.907 | 0.906 | 0.900 | 0.883 |
| 3 | 0.903 | 0.943 | 0.971 | 0.946 | 0.778 |
| 4 | 0.904 | 0.944 | 0.972 | 0.947 | 0.777 |
| 5 | 0.887 | 0.926 | 0.954 | 0.926 | 0.758 |

Table 5: Duo heuristic results for the spatial join task. The best duo heuristic configuration is parallel (min_angle=10) and overlap (max_area=0.2/0.3).

|  overlap (max_area=0.1) | | | | |
|---|---|---|---|---|
| **parallel** | **clearance (min_distance)** | | | |
| **(min_angle)** | 1 | 2 | 3 | 4 | 5 |
| 1 | 0.884 | 0.884 | 0.882 | 0.881 | 0.863 |
| 2 | 0.932 | 0.932 | 0.930 | 0.929 | 0.907 |
| 5 | 0.972 | 0.972 | 0.970 | 0.969 | 0.946 |
| 10 | 0.977 | 0.977 | 0.975 | 0.974 | 0.951 |
| 20 | 0.969 | 0.969 | 0.967 | 0.966 | 0.943 |

|  overlap (max_area=0.2) | | | | |
|---|---|---|---|---|
| **parallel** | **clearance (min_distance)** | | | |
| **(min_angle)** | 1 | 2 | 3 | 4 | 5 |
| 1 | 0.888 | 0.888 | 0.886 | 0.885 | 0.867 |
| 2 | 0.937 | 0.937 | 0.935 | 0.934 | 0.912 |
| 5 | 0.980 | 0.980 | 0.978 | 0.977 | 0.954 |
| 10 | 0.987 | 0.987 | 0.985 | 0.984 | 0.961 |
| 20 | 0.985 | 0.985 | 0.983 | 0.982 | 0.959 |

|  overlap (max_area=0.3) | | | | |
|---|---|---|---|---|
| **parallel** | **clearance (min_distance)** | | | |
| **(min_angle)** | 1 | 2 | 3 | 4 | 5 |
| 1 | 0.887 | 0.887 | 0.885 | 0.884 | 0.866 |
| 2 | 0.936 | 0.936 | 0.934 | 0.933 | 0.911 |
| 5 | 0.978 | 0.978 | 0.976 | 0.975 | 0.952 |
| 10 | 0.987 | 0.987 | 0.985 | 0.984 | 0.961 |
| 20 | 0.990 | 0.990 | 0.988 | 0.987 | 0.964 |

|  overlap (max_area=0.4) | | | | |
|---|---|---|---|---|
| **parallel** | **clearance (min_distance)** | | | |
| **(min_angle)** | 1 | 2 | 3 | 4 | 5 |
| 1 | 0.862 | 0.862 | 0.860 | 0.859 | 0.842 |
| 2 | 0.904 | 0.904 | 0.902 | 0.901 | 0.880 |
| 5 | 0.942 | 0.942 | 0.940 | 0.939 | 0.917 |
| 10 | 0.951 | 0.951 | 0.949 | 0.948 | 0.926 |
| 20 | 0.953 | 0.953 | 0.951 | 0.950 | 0.928 |

|  overlap (max_area=0.5) | | | | |
|---|---|---|---|---|
| **parallel** | **clearance (min_distance)** | | | |
| **(min_angle)** | 1 | 2 | 3 | 4 | 5 |
| 1 | 0.733 | 0.733 | 0.731 | 0.730 | 0.715 |
| 2 | 0.763 | 0.763 | 0.761 | 0.760 | 0.741 |
| 5 | 0.779 | 0.779 | 0.777 | 0.776 | 0.756 |
| 10 | 0.781 | 0.781 | 0.779 | 0.778 | 0.758 |
| 20 | 0.781 | 0.781 | 0.779 | 0.778 | 0.758 |

Table 6: Trio heuristic results for the spatial join task. The best trio heuristic configuration is parallel (min_angle=20), overlap (max_area=0.3) and clearance (min_distance=1/2).

## C.2 Spatial Union Heuristic Results

| **parallel** **(min_angle)** | **Accuracy** **(acc)** |
|---|---|
| 1 | 0.857 |
| 2 | 0.867 |
| 3 | 0.857 |
| 4 | 0.842 |
| 5 | 0.842 |

| **overlap** **(max_area)** | **Accuracy** **(acc)** |
|---|---|
| 0.5 | 0.882 |
| 0.6 | 0.902 |
| 0.7 | 0.910 |
| 0.8 | 0.927 |
| 0.9 | 0.917 |

Table 7: Single heuristic results for the spatial union task. The best single heuristic configuration is parallel (max_area=0.8).

| **parallel** | **overlap (max_area)** | | | | |
|---|---|---|---|---|---|
| **(min_angle)** | 0.5 | 0.6 | 0.7 | 0.8 | 0.9 |
| 1 | 0.922 | 0.910 | 0.905 | 0.902 | 0.870 |
| 2 | 0.955 | 0.945 | 0.940 | 0.937 | 0.902 |
| 3 | 0.962 | 0.952 | 0.950 | 0.945 | 0.912 |
| 4 | 0.952 | 0.947 | 0.947 | 0.945 | 0.912 |
| 5 | 0.955 | 0.950 | 0.950 | 0.947 | 0.915 |

Table 8: Duo heuristic results for the spatial union task. The best duo heuristic configuration is parallel (min_angle=3) and overlap (max_area=0.5)

# D    Heuristic-Driven Prompting Results

| prompting method | heuristic information | heuristic | models | | | | |
|---|---|---|---|---|---|---|---|
| | | | llama3 | mistral | 4o-mini | qwen-plus | 4o |
| zero-shot | plain | - | 0.520 | 0.598 | 0.405 | 0.603 | 0.580 |
| | hints | parallel (p) | 0.520 | 0.600 | 0.608 | 0.645 | 0.546 |
| | | clearance (c) | 0.518 | 0.599 | 0.516 | 0.609 | 0.394 |
| | | overlap (o) | 0.528 | 0.599 | 0.596 | 0.603 | 0.549 |
| | | (p,c) | 0.493 | 0.599 | 0.479 | 0.676 | 0.399 |
| | | (p,o) | 0.505 | 0.600 | 0.458 | 0.669 | 0.413 |
| | | (c,o) | 0.536 | 0.599 | 0.402 | 0.633 | 0.410 |
| | | (p,c,o) | 0.556 | 0.599 | 0.400 | 0.705 | 0.406 |
| | values | (p) | 0.575 | 0.675 | 0.938 | 0.946 | 0.948 |
| | | (c) | 0.596 | 0.599 | 0.742 | 0.732 | 0.670 |
| | | (o) | 0.570 | 0.599 | 0.601 | 0.601 | 0.696 |
| | | (p,c) | 0.523 | 0.599 | 0.877 | 0.950 | 0.948 |
| | | (p,o) | 0.561 | 0.599 | 0.664 | 0.949 | 0.974 |
| | | (c,o) | 0.551 | 0.604 | 0.734 | 0.733 | 0.529 |
| | | (p,c,o) | 0.563 | 0.599 | 0.679 | 0.949 | 0.861 |
| few-shot | plain | - | 0.495 | 0.450 | 0.495 | 0.633 | 0.629 |
| | hints | (p) | 0.472 | 0.522 | 0.589 | 0.682 | 0.628 |
| | | (c) | 0.526 | 0.668 | 0.512 | 0.679 | 0.589 |
| | | (o) | 0.486 | 0.618 | 0.607 | 0.644 | 0.636 |
| | | (p,c) | 0.516 | 0.579 | 0.548 | 0.663 | 0.606 |
| | | (p,o) | 0.459 | 0.559 | 0.575 | 0.655 | 0.591 |
| | | (c,o) | 0.474 | 0.652 | 0.590 | 0.654 | 0.483 |
| | | (p,c,o) | 0.508 | 0.611 | 0.514 | 0.653 | 0.565 |
| | values | (p) | 0.606 | 0.867 | 0.929 | 0.936 | 0.931 |
| | | (c) | 0.485 | 0.597 | 0.822 | 0.845 | 0.837 |
| | | (o) | 0.505 | 0.657 | 0.760 | 0.687 | 0.727 |
| | | (p,c) | 0.517 | 0.676 | 0.916 | 0.941 | 0.940 |
| | | (p,o) | 0.479 | 0.618 | 0.858 | 0.968 | 0.959 |
| | | (c,o) | 0.490 | 0.457 | 0.906 | 0.948 | 0.939 |
| | | (p,c,o) | 0.539 | 0.625 | 0.872 | 0.984 | 0.967 |

Table 9: Accuracy of heuristic-driven prompting method for the spatial join task. We evaluate both zero-shot and few-shot prompting, with plain task description, heuristic-hints or heuristic features.

| prompting method | heuristic information | heuristic | models | | | | |
|---|---|---|---|---|---|---|---|
| | | | llama3 | mistral | 4o-mini | qwen-plus | 4o |
| zero-shot | plain | - | 0.436 | 0.366 | 0.454 | 0.799 | 0.817 |
| | hints | parallel (p) | 0.544 | 0.43 | | | |
| | | overlap (o) | 0.506 | 0.619 | 0.481 | 0.852 | 0.777 |
| | | (p,o) | 0.476 | 0.624 | 0.414 | 0.752 | 0.524 |
| | values | (p) | 0.622 | 0.639 | 0.777 | 0.817 | 0.837 |
| | | (o) | 0.549 | 0.579 | 0.744 | 0.865 | 0.862 |
| | | (p,o) | 0.566 | 0.632 | 0.797 | 0.927 | 0.955 |
| few-shot | plain | - | 0.414 | 0.361 | 0.521 | 0.835 | 0.802 |
| | hints | (p) | 0.383 | 0.363 | 0.629 | 0.822 | 0.777 |
| | | (o) | 0.391 | 0.421 | 0.647 | 0.827 | 0.779 |
| | | (p,o) | 0.373 | 0.424 | 0.564 | 0.762 | 0.815 |
| | values | (p) | 0.491 | 0.792 | 0.769 | 0.835 | 0.820 |
| | | (o) | 0.456 | 0.471 | 0.882 | 0.910 | 0.890 |
| | | (p,o) | 0.466 | 0.779 | 0.805 | 0.960 | 0.882 |

Table 10: Accuracy of heuristic-driven prompting method for the spatial union task. We evaluate both zero-shot and few-shot prompting, with plain task description, heuristic-hints or heuristic features.

# E   Mistake Examples

Table 11 presents the common mistakes made by the three models on the spatial join task.

The **Wrong Logic** category describes logical mistakes made by the models when analyzing spatial relationships between the paired objects. For example, when calculating the distance between the sidewalk and the road, using the first point of road and last point of sidewalk is incorrect.

**Unclear Calculation** describes mistakes where there is no explicit demonstration by the models of the calculation steps used to arrive at an answer, though some numerical evidence may be provided. In such cases, models tend to use words such as "appears" and "approximately" rather than fully demonstrating their computations, casting uncertainty upon whether the models actually execute the calculation, and if so, how they do so.

The **No Calculation** category is similar to the **Unclear Calculation** category. The difference is that, in this category, the model does not provide any numerical evidence whatsoever for its answers. The models tend to verbalize their assumptions, which they use to arrive at a final conclusion for a task.

| common mistakes | examples |
|---|---|
| **wrong logic** | - orientation of road/sidewalk with first and last point |
| | - distance calculation with first point of road and last point of sidewalk |
| **unclear calculation** | - the closest approach between them appears to be around 2-3 meters |
| | the closest sidewalk point to the road is approximately 5-10 meters away |
| | - the distance is approximately 3 meters |
| **no calculation** | - let's assume the calculations have been performed |
| | - let's assume the computed distance is approximately 1 meter |
| | - the sidewalk appears to be running parallel |

Table 11: Common mistake examples in the reasoning steps from {4o-mini, qwen-plus, 4o}.

## F Prompting Examples

---

**Spatial Join, Zero-Shot (plain)**

**System:**
### Instruction:
You are a helpful geospatial analysis assistant. I will provide you with a pair of (sidewalk, road) geometries in GeoJSON format. Your task is to determine whether the sidewalk runs alongside the road. If it does, return 1. Otherwise, return 0. No explanation is needed.

**User:**
### Input:
Sidewalk: "coordinates": [[-122.20182090000002, 47.630587], [-122.2018092, 47.630594599999995], [-122.2018007, 47.6306025], [-122.2017952, 47.63061540000001], [-122.20179480000002, 47.63063060000001], [-122.20179549999999, 47.6309796]], "type": "LineString"
Road: "coordinates": [[-122.2018393, 47.6305377], [-122.2016519, 47.6305356]], "type": "LineString"

**Assistant:**
### Response: {}

---

Table 12: Zero-shot, no heuristics (plain) prompt for the spatial join task.

---

**Spatial Join, Zero-Shot (hints)**

**System:**
### Instruction:
You are a helpful geospatial analysis assistant. I will provide you with a pair of (sidewalk, road) geometries in GeoJSON format. Your task is to determine whether the sidewalk runs alongside the road by evaluating the following conditions:

- Parallelism: The sidewalk should be approximately parallel to the road, with only a small angle difference between their orientations.
- Clearance: The sidewalk and road must not overlap or intersect, and they must maintain a certain distance apart.
- Overlap: The sidewalk and road must not directly overlap, but a 10-meter buffer around each should have a certain amount of overlap.

If all conditions are satisfied, return 1. Otherwise, return 0. No explanation is needed.

**User:**
### Input:
Sidewalk: "coordinates": [[-122.20182090000002, 47.630587], [-122.2018092, 47.630594599999995], [-122.2018007, 47.6306025], [-122.2017952, 47.63061540000001], [-122.20179480000002, 47.63063060000001], [-122.20179549999999, 47.6309796]], "type": "LineString"
Road: "coordinates": [[-122.2018393, 47.6305377], [-122.2016519, 47.6305356]], "type": "LineString"

**Assistant:**
### Response: {}

---

Table 13: Zero-shot, with heuristic hints prompt for the spatial join task. In this prompt, we provide all three heuristic hints. For prompt with single and duo heuristic hints, the same instruction is used but with only one or two heuristic hints.

---

**Spatial Join, Zero-Shot (features)**

**System:**
### Instruction:
You are a helpful geospatial analysis assistant. I will provide you with a pair of (sidewalk, road) geometries in GeoJSON format, along with three key statistics:

- min_angle: The minimum angle (in degrees) between the sidewalk and the road.
- min_distance: The minimum distance (in meters) between the sidewalk and the road.
- max_area: The maximum percentage of overlapping area relative to the sidewalk and road, considering a 10-meter buffer.

Your task is to determine whether the sidewalk runs alongside the road by evaluating the following conditions:

- Parallelism: The sidewalk should be approximately parallel to the road, with only a small angle difference between their orientations. The min_angle value provides a measure of this alignment.
- Clearance: The sidewalk and road must not overlap or intersect, and they must maintain a certain distance apart. The min_distance value helps quantify this proximity.
- Overlap: The sidewalk and road must not directly overlap, but a 10-meter buffer around each should have a certain amount of overlap. The max_area values help quantify this overlap and should not be near zero or too small.

If all conditions are satisfied, return 1. Otherwise, return 0. No explanation is needed.

**User:**
### Input:
Sidewalk: "coordinates": [[-122.20182090000002, 47.630587], [-122.2018092, 47.630594599999995], [-122.2018007, 47.6306025], [-122.2017952, 47.63061540000001], [-122.20179480000002, 47.63063060000001], [-122.20179549999999, 47.6309796]], "type": "LineString"
Road: "coordinates": [[-122.2018393, 47.6305377], [-122.2016519, 47.6305356]], "type": "LineString"
min_angle: 1.094690596035889
min_distance: 8.916691213846207
max_area: 0.538494352036627

**Assistant:**
### Response: {}

---

Table 14: Zero-shot, with heuristic features prompt for the spatial join task. In this prompt, we provide all three heuristic features. For prompt with single and duo heuristic features, the same instruction is used but with only one or two heuristic features.

---

**Spatial Join, Few-Shot (plain)**

**System:**
### Instruction:
You are a helpful geospatial analysis assistant. I will provide you with a pair of (sidewalk, road) geometries in GeoJSON format. Your task is to determine whether the sidewalk runs alongside the road. If it does, return 1. Otherwise, return 0. No explanation is needed.

### First Example:
Sidewalk: 'coordinates': [[-122.15646960000001, 47.58741259999999], [-122.1562564, 47.58744089999999]], "type": "LineString"
Road: 'coordinates': [[-122.1563888, 47.5874271], [-122.1563897, 47.5874341], [-122.1564949, 47.5890663], [-122.1564975, 47.5890982]], "type": "LineString"
Response: {0}

### Second Example:
Sidewalk: 'coordinates': [[-122.13341579999998, 47.54698270000001], [-122.1334011, 47.5468383]], "type": "LineString"
Road: 'coordinates': [[-122.1328993, 47.5458957], [-122.1329478, 47.5460104], [-122.1330183, 47.5461317], [-122.1330885, 47.5462402], [-122.1333795, 47.5466214], [-122.1334411, 47.5467369], [-122.1334757, 47.5468199], [-122.1335148, 47.5469582]], "type": "LineString"
Response: {1}

**User:**
### Input:
Sidewalk: "coordinates": [[-122.20182090000002, 47.630587], [-122.2018092, 47.630594599999995], [-122.2018007, 47.6306025], [-122.2017952, 47.63061540000001], [-122.20179480000002, 47.63063060000001], [-122.20179549999999, 47.6309796]], "type": "LineString"
Road: "coordinates": [[-122.2018393, 47.6305377], [-122.2016519, 47.6305356]], "type": "LineString"

**Assistant:**
### Response: {}

Table 15: Few-shot, no heuristics (plain) prompt for the spatial join task.

---

**Spatial Join, Zero-Shot (hints)**

**System:**
### Instruction:
You are a helpful geospatial analysis assistant. I will provide you with a pair of (sidewalk, road) geometries in GeoJSON format. Your task is to determine whether the sidewalk runs alongside the road by evaluating the following conditions:

- Parallelism: The sidewalk should be approximately parallel to the road, with only a small angle difference between their orientations.
- Clearance: The sidewalk and road must not overlap or intersect, and they must maintain a certain distance apart.
- Overlap: The sidewalk and road must not directly overlap, but a 10-meter buffer around each should have a certain amount of overlap.

If all conditions are satisfied, return 1. Otherwise, return 0. No explanation is needed.

### First Example:
Sidewalk: 'coordinates': [[-122.15646960000001, 47.58741259999999], [-122.1562564, 47.58744089999999]], "type": "LineString"
Road: 'coordinates': [[-122.1563888, 47.5874271], [-122.1563897, 47.5874341], [-122.1564949, 47.5890663], [-122.1564975, 47.5890982]], "type": "LineString"
Response: {0}

### Second Example:
Sidewalk: 'coordinates': [[-122.13341579999998, 47.54698270000001], [-122.1334011, 47.5468383]], "type": "LineString"
Road: 'coordinates': [[-122.1328993, 47.5458957], [-122.1329478, 47.5460104], [-122.1330183, 47.5461317], [-122.1330885, 47.5462402], [-122.1333795, 47.5466214], [-122.1334411, 47.5467369], [-122.1334757, 47.5468199], [-122.1335148, 47.5469582]], "type": "LineString"
Response: {1}

**User:**
### Input:
Sidewalk: "coordinates": [[-122.20182090000002, 47.630587], [-122.2018092, 47.630594599999995], [-122.2018007, 47.6306025], [-122.2017952, 47.63061540000001], [-122.20179480000002, 47.63063060000001], [-122.20179549999999, 47.6309796]], "type": "LineString"
Road: "coordinates": [[-122.2018393, 47.6305377], [-122.2016519, 47.6305356]], "type": "LineString"

**Assistant:**
### Response: {}

Table 16: Few-shot, with heuristic hints prompt for the spatial join task. In this prompt, we provide all three heuristic hints. For prompt with single and duo heuristic hints, the same instruction is used but with only one or two heuristic hints.

---

**Spatial Join, Zero-Shot (features)**

**System:**
### Instruction:
You are a helpful geospatial analysis assistant. I will provide you with a pair of (sidewalk, road) geometries in GeoJSON format, along with three key statistics:

- min_angle: The minimum angle (in degrees) between the sidewalk and the road.
- min_distance: The minimum distance (in meters) between the sidewalk and the road.
- max_area: The maximum percentage of overlapping area relative to the sidewalk and road, considering a 10-meter buffer.

Your task is to determine whether the sidewalk runs alongside the road by evaluating the following conditions:

- Parallelism: The sidewalk should be approximately parallel to the road, with only a small angle difference between their orientations. The min_angle value provides a measure of this alignment.
- Clearance: The sidewalk and road must not overlap or intersect, and they must maintain a certain distance apart. The min_distance value helps quantify this proximity.
- Overlap: The sidewalk and road must not directly overlap, but a 10-meter buffer around each should have a certain amount of overlap. The max_area values help quantify this overlap and should not be near zero or too small.

If all conditions are satisfied, return 1. Otherwise, return 0. No explanation is needed.

### First Example:
Sidewalk: 'coordinates': [[-122.15646960000001, 47.58741259999999], [-122.1562564, 47.58744089999999]], "type": "LineString"
Road: 'coordinates': [[-122.1563888, 47.5874271], [-122.1563897, 47.5874341], [-122.1564949, 47.5890663], [-122.1564975, 47.5890982]], "type": "LineString"
min_angle: 86.12658269425465
min_distance: 0.6112785794641761
max_area: 0.4346838047603181
Response: {0}

### Second Example:
Sidewalk: 'coordinates': [[-122.13341579999998, 47.546982700000001], [-122.1334011, 47.5468383]], "type": "LineString"
Road: 'coordinates': [[-122.1328993, 47.5458957], [-122.1329478, 47.5460104], [-122.1330183, 47.5461317], [-122.1330885, 47.5462402], [-122.1333795, 47.5466214], [-122.1334411, 47.5467369], [-122.1334757, 47.5468199], [-122.1335148, 47.5469582]], "type": "LineString"
min_angle: 9.973873687169487
min_distance: 8.72605420848234
max_area: 0.4654527079273675
Response: {1}

**User:**
### Input:
Sidewalk: "coordinates": [[-122.20182090000002, 47.630587], [-122.2018092, 47.630594599999995], [-122.2018007, 47.6306025], [-122.2017952, 47.63061540000001], [-122.20179480000002, 47.63063060000001], [-122.20179549999999, 47.6309796]], "type": "LineString"
Road: "coordinates": [[-122.2018393, 47.6305377], [-122.2016519, 47.6305356]], "type": "LineString"
min_angle: 1.094690596035889
min_distance: 8.916691213846207
max_area: 0.538494352036627

**Assistant:**
### Response: {}

---

Table 17: Few-shot, with heuristic features prompt for the spatial join task. In this prompt, we provide all three heuristic features. For prompt with single and duo heuristic features, the same instruction is used but with only one or two heuristic features.

---

**Spatial Union, Zero-Shot (plain)**

**System:**
### Instruction:
You are a helpful geospatial analysis assistant. I will provide you with a pair of (sidewalk 1, sidewalk 2) geometries in GeoJSON format. Your task is to determine whether these two geometries represent the same sidewalk, either fully or partially. If they do, return 1. Otherwise, return 0. No explanation is needed.

**User:**
### Input:
Sidewalk: "coordinates": [[-122.20182090000002, 47.630587], [-122.2018092, 47.630594599999995], [-122.2018007, 47.6306025], [-122.2017952, 47.63061540000001], [-122.20179480000002, 47.63063060000001], [-122.20179549999999, 47.6309796]], "type": "LineString"
Road: "coordinates": [[-122.2018393, 47.6305377], [-122.2016519, 47.6305356]], "type": "LineString"

**Assistant:**
### Response: {}

---

Table 18: Zero-shot, no heuristics (plain) prompt for the spatial union task.

---

**Spatial Union, Zero-Shot (hints)**

**System:**
### Instruction:
You are a helpful geospatial analysis assistant. I will provide you with a pair of (sidewalk 1, sidewalk 2) geometries in GeoJSON format. Your task is to determine whether these two geometries represent the same sidewalk, either fully or partially, by evaluating the following conditions:

- Parallelism: The two sidewalks should be approximately parallel, with only a small angular difference in their orientations.
- Overlap: The two sidewalks must fully or partially overlap. Simply connecting at the endpoints does not count as an intersection.

If all conditions are satisfied, return 1. Otherwise, return 0. No explanation is needed.

**User:**
### Input:
Sidewalk: "coordinates": [[-122.20182090000002, 47.630587], [-122.2018092, 47.630594599999995], [-122.2018007, 47.6306025], [-122.2017952, 47.63061540000001], [-122.20179480000002, 47.63063060000001], [-122.20179549999999, 47.6309796]], "type": "LineString"
Road: "coordinates": [[-122.2018393, 47.6305377], [-122.2016519, 47.6305356]], "type": "LineString"

**Assistant:**
### Response: {}

---

Table 19: Zero-shot, with heuristic hints prompt for the spatial union task. In this prompt, we provide duo heuristic hints. For prompt with single heuristic hint, the same instruction is used but with only one heuristic hint.

---

**Spatial Union, Zero-Shot (features)**

**System:**
### Instruction:
You are a helpful geospatial analysis assistant. I will provide you with a pair of (sidewalk, road) geometries in GeoJSON format, along with two key statistics:

- min_angle: The minimum angle (in degrees) between the sidewalk and the road.
- max_area: The maximum percentage of overlapping area relative to the sidewalk and road, considering a 10-meter buffer.

Your task is to determine whether these two geometries represent the same sidewalk, either fully or partially, by evaluating the following conditions:

- Parallelism: The two sidewalks should be approximately parallel, with only a small angular difference in their orientations. The min_angle value provides a measure of this alignment.
- Overlap: The two sidewalks must fully or partially overlap. Simply connecting at the endpoints does not count as an intersection. The max_area values help quantify this overlap.

If both conditions are satisfied, return 1. Otherwise, return 0. No explanation is needed.

**User:**
### Input:
Sidewalk: "coordinates": [[-122.20182090000002, 47.630587], [-122.2018092, 47.630594599999995], [-122.2018007, 47.6306025], [-122.2017952, 47.63061540000001], [-122.20179480000002, 47.63063060000001], [-122.20179549999999, 47.6309796]], "type": "LineString"
Road: "coordinates": [[-122.2018393, 47.6305377], [-122.2016519, 47.6305356]], "type": "LineString"
min_angle: 1.094690596035889
min_distance: 8.916691213846207
max_area: 0.538494352036627

**Assistant:**
### Response: {}

---

Table 20: Zero-shot, with heuristic features prompt for the spatial union task. In this prompt, we provide duo heuristic features. For prompt with single heuristic feature, the same instruction is used but with only one heuristic feature.

---

**Spatial Union, Few-Shot (plain)**

**System:**
### Instruction:
You are a helpful geospatial analysis assistant. I will provide you with a pair of (sidewalk 1, sidewalk 2) geometries in GeoJSON format. Your task is to determine whether these two geometries represent the same sidewalk, either fully or partially. If they do, return 1. Otherwise, return 0. No explanation is needed.

### First Example:
Sidewalk 1: 'coordinates': [[-122.1731058, 47.5709015], [-122.1743964, 47.5709069]], 'type': 'LineString'
Sidewalk 2: 'coordinates': [[-122.1743856, 47.570905], [-122.1743848, 47.5710029]], 'type': 'LineString'
Response: {0}

### Second Example:
Sidewalk 1: 'coordinates': [[-122.17248400000001, 47.570692699999995], [-122.17136010000002, 47.5706601], [-122.1698608, 47.5706113]], 'type': 'LineString'
Sidewalk 2: 'coordinates': [[-122.1700114, 47.5706215], [-122.1703076, 47.5706256]], 'type': 'LineString'
Response: {1}

**User:**
### Input:
Sidewalk: "coordinates": [[-122.20182090000002, 47.630587], [-122.2018092, 47.630594599999995], [-122.2018007, 47.6306025], [-122.2017952, 47.63061540000001], [-122.20179480000002, 47.63063060000001], [-122.20179549999999, 47.6309796]], "type": "LineString"
Road: "coordinates": [[-122.2018393, 47.6305377], [-122.2016519, 47.6305356]], "type": "LineString"

**Assistant:**
### Response: {}

Table 21: Few-shot, no heuristics (plain) prompt for the spatial union task.

---

**Spatial Union, Few-Shot (hints)**

**System:**
### Instruction:
You are a helpful geospatial analysis assistant. I will provide you with a pair of (sidewalk, road) geometries in GeoJSON format. Your task is to determine whether the sidewalk runs alongside the road by evaluating the following conditions:

- Parallelism: The sidewalk should be approximately parallel to the road, with only a small angle difference between their orientations.
- Clearance: The sidewalk and road must not overlap or intersect, and they must maintain a certain distance apart.
- Overlap: The sidewalk and road must not directly overlap, but a 10-meter buffer around each should have a certain amount of overlap.

If all conditions are satisfied, return 1. Otherwise, return 0. No explanation is needed.

### First Example:
Sidewalk 1: 'coordinates': [[-122.1731058, 47.5709015], [-122.1743964, 47.5709069]], 'type': 'LineString'
Sidewalk 2: 'coordinates': [[-122.1743856, 47.570905], [-122.1743848, 47.5710029]], 'type': 'LineString'
Response: {0}

### Second Example:
Sidewalk 1: 'coordinates': [[-122.17248400000001, 47.570692699999995], [-122.17136010000002, 47.5706601], [-122.1698608, 47.5706113]], 'type': 'LineString'
Sidewalk 2: 'coordinates': [[-122.1700114, 47.5706215], [-122.1703076, 47.5706256]], 'type': 'LineString'
Response: {1}

**User:**
### Input:
Sidewalk: "coordinates": [[-122.20182090000002, 47.630587], [-122.2018092, 47.630594599999995], [-122.2018007, 47.6306025], [-122.2017952, 47.63061540000001], [-122.20179480000002, 47.63063060000001], [-122.20179549999999, 47.6309796]], "type": "LineString"
Road: "coordinates": [[-122.2018393, 47.6305377], [-122.2016519, 47.6305356]], "type": "LineString"

**Assistant:**
### Response: {}

Table 22: Few-shot, with heuristic hints prompt for the spatial union task. In this prompt, we provide duo heuristic hints. For prompt with single heuristic hint, the same instruction is used but with only one heuristic hint.

---

**Spatial Union, Few-Shot (features)**

**System:**
### Instruction:
You are a helpful geospatial analysis assistant. I will provide you with a pair of (sidewalk, road) geometries in GeoJSON format, along with three key statistics:

- min_angle: The minimum angle (in degrees) between the sidewalk and the road.
- min_distance: The minimum distance (in meters) between the sidewalk and the road.
- max_area: The maximum percentage of overlapping area relative to the sidewalk and road, considering a 10-meter buffer.

Your task is to determine whether the sidewalk runs alongside the road by evaluating the following conditions:

- Parallelism: The sidewalk should be approximately parallel to the road, with only a small angle difference between their orientations.
- Clearance: The sidewalk and road must not overlap or intersect, and they must maintain a certain distance apart.
- Overlap: The sidewalk and road must not directly overlap, but a 10-meter buffer around each should have a certain amount of overlap.

If all conditions are satisfied, return 1. Otherwise, return 0. No explanation is needed.

### First Example:
Sidewalk 1: 'coordinates': [[-122.1731058, 47.5709015], [-122.1743964, 47.5709069]], 'type': 'LineString'
Sidewalk 2: 'coordinates': [[-122.1743856, 47.570905], [-122.1743848, 47.5710029]], 'type': 'LineString'
min_angle: 89.77154191724516
max_area: 0.5473559863066643
Response: {0}

### Second Example:
Sidewalk 1: 'coordinates': [[-122.17248400000001, 47.570692699999995], [-122.17136010000002, 47.5706601], [-122.1698608, 47.5706113]], 'type': 'LineString'
Sidewalk 2: 'coordinates': [[-122.1700114, 47.5706215], [-122.1703076, 47.5706256]], 'type': 'LineString'
min_angle: 0.8684260514779112
max_area: 0.9832547250026938
Response: {1}

### Input:
Sidewalk 1: "coordinates": [[-122.1934421, 47.6172096], [-122.19246919999999, 47.61719890000001], [-122.19217600000002, 47.6171954], [-122.1912306, 47.61719189999999], [-122.19120270000002, 47.6171917]], "type": "LineString"
Sidewalk 2: "coordinates": [[-122.19217, 47.6171059], [-122.1921715, 47.617199]], "type": "LineString"
min_angle: 88.39302577789853
max_area: 0.5886569923778128

**User:**
### Input:
Sidewalk: "coordinates": [[-122.20182090000002, 47.630587], [-122.2018092, 47.630594599999995], [-122.2018007, 47.6306025], [-122.2017952, 47.63061540000001], [-122.20179480000002, 47.63063060000001], [-122.20179549999999, 47.6309796]], "type": "LineString"
Road: "coordinates": [[-122.2018393, 47.6305377], [-122.2016519, 47.6305356]], "type": "LineString"
min_angle: 1.094690596035889
min_distance: 8.916691213846207
max_area: 0.538494352036627

**Assistant:**
### Response: {}

---

Table 23: Few-shot, with heuristic features prompt for the spatial union task. In this prompt, we provide duo heuristic features. For prompt with single heuristic feature, the same instruction is used but with only one heuristic feature.

