# OpenReview forum: "Can Large Language Models Integrate Spatial Data? Empirical Insights into Reasoning Strengths and Computational Weaknesses"
_colmweb.org/COLM/2025/Conference — COLM 2025_

### Official Review · Reviewer_ap53 · 2025-05-05

**Rating:** 5
**Confidence:** 4
**Ethics Flag:** 1

**Summary:**

This work studies the usage of LLMs to integrate spatial data. The tasks studied are simplified, coordinate representations of start and end points of sidewalks and roads, and the tasks are to binarily classify for either union or join. The authors release a dataset for both subtasks, and then experiment with heuristics and LLMs. They find that heuristics can perform well given some threshold tuning and feature selection. They find that LLMs when prompted with coordinates alone underperform, but when prompted with the heuristics can do well, and even better with a two-stage review and refine. The work also performs some qualitative analysis of LLM's "reasoning" outputs to show that while they mention relevant heuristics, the reasoning can be wrong.

**Questions To Authors:**

* Table 1: can you include in the appendix the exact configurations that were the best and worst?  For example, something like "trio used parallelism 5, clearance 3, overlap 0.4)
* please drop the "plain" settings, as underspecifying the instructions is not interesting. Use "hints" as the simplest baseline.
* Figure 3: where does "avg heuristics" come from? Consider the Union chart; avg heuristics is .918, but all of the other cells are well under that.
* Figure 4: there is the same title for both charts, right should be Union
* Can you add a discussion on why review & refine achieves such great performance (99% for join, 94% for union) as compared to the heuristic prompting (90%, 92%)

**Reasons To Accept:**

* The writing is generally clear, and the paper is organized well.
* Experimentation is fairly comprehensive, covering 5 LLMs, a reasoning model, and 2 subtasks.
* The qualitative analysis in 5.3 is interesting, as it shows the LLMs mention spatial conditions of proximity, distance and angle, even if their answers are wrong. There is also qualitative analysis in the other sections, which is always good to have.

**Reasons To Reject:**

* My main concern is that the task studied, on coordinates for sidewalks in either join or union subtasks, is limited in scope, and simple to perform. This means that using benchmarking LLMs on them is not well motivated, and the task as-is is feels like a toy task. In other words, I think COLM may not be the best venue for the current framing of the paper.
    * First, we see that the heuristic approaches based on 1-3 features can already achieve >90% accuracy (Table 1). I get that finding an optimal threshold for each heuristic takes some effort in "feature selection and threshold tuning.", but this is a one-time effort for each subtask. Compared to LLMs, where you needed to prompt engineer as well as run costly inference for each example.
    * Second, Figure 2 shows that the tasks can easily be solved by visual inspection. In fact, many open and proprietary LLMs in 2025 can support visual input. I think this would be another very strong baseline, using the visualizations as input. Using 2 coordinates as textual input can hardly be considered a thorough analysis of "spatial data", just a small part of it.
* It is unclear how the data was split into train validation and test sets, after reading Appendix A. Is it only random, or was it stratified? If it's only random, there could be possible leakage between splits, as annotations may overlap in the roads or coordinates.
* The results of T2 are rather obvious. LLMs are known to not be great at spatial reasoning, and for the second part, obviously heuristics would help LLMs if they can directly solve the tasks alone.

---

> ### Author Response · Authors · 2025-06-02
> **Rebuttal by Authors**
>
> We appreciate the reviews’ thoughtful questions. Here are our responses:
>
> > Include the best and worst exact heuristic configurations.
>
> Here are the configurations and we will include them in the appendix.
>
> - Spatial join – best single (p=10); best duo (p=10, o=0.2); best trio (p=20, o=0.3, c=2); worst single (o=0.5); worst duo (p=1, o=0.5); worst trio (p=1, o=0.5, c=5).
> - Spatial union: – best single (o=0.8); best duo (p=3, o=0.5); worst single (p=4); worst duo (p=1, o=0.9).
>
> > Drop the "plain" settings. Use "hints" as the simplest baseline.
>
> We agree that dropping the “plain” setting makes sense given its poor performance. Even with hints, models struggle with the tasks. We will use the “hints” as our simplest baseline.
>
> > Figure 3: where does "avg heuristics" come from?
>
> The “avg heuristics” is the average performance across all heuristic combinations (single, duo, trio) and remains constant across all LLM prompting settings for comparison. We will update the visualization to clarify this.
>
> > Figure 4: there is the same title for both charts.
>
> Thank you for the careful reading. We will fix the title.
>
> > Why review & refine is better compared to the heuristic prompting
>
> 1. review-and-refine introduces additional reasoning steps that enable deeper analysis and better decision-making.
>
> 2. Initial answers affect results; for instance, qwen-plus’s refined outcomes range from 94.9% for weaker priors up to 98.4% for stronger ones. This shows review-and-refine benefits from iterative reasoning and good initial guidance.
>
> > The task is too simple and narrow, making it unsuitable for the current venue.
>
> We appreciate these concerns, but hope to convince you that our work is well-motivated and is a good fit for COLM:
>
> 1.  **Our work expands the existing understanding of LLM capabilities to encompass spatial data with real-world geometry objects.** While the two tasks may appear simple, they are foundational to more complex spatial analysis. Our results show that even these simple tasks challenge LLMs, revealing computational limitations.
>
> 2. **We examined the use of LLMs in urban spatial data integration — an underexplored area with meaningful real-world impact, especially in accessibility.** We see our work as a step toward enabling LLM-driven systems to unify fragmented spatial data, supporting more accurate representations of infrastructure like sidewalks and intersections for accessible urban navigation.
>
> 3. We certainly agree that evaluating a wider range of spatial tasks can provide deeper and more general insight into LLMs’ capabilities in this domain. In response, we conducted additional experiments on four more spatial analysis tasks involving spatial reasoning and geometric computation. **Due to space constraints, we have included these results in our response to Reviewer VhgA's second question, and hope that you’ll take the time to examine them.**
>
> > Simple heuristics with minimal tuning are more efficient than LLMs that require prompt engineering and inference cost.
>
> **Feature selection and threshold tuning are not one-time efforts; rather, they vary across locations (e.g., cities with distinct physical layouts) and temporal changes (e.g., road expansions).** This limits the generalizability of heuristic-based methods. While LLM performance is also context-sensitive, LLMs offer the advantage of integrating external information—such as city-specific context or visual data—and drawing on broad world knowledge, potentially enabling better adaptation across diverse scenarios. Though LLMs can be costly, they may ultimately reduce the manual effort required to tailor spatial analysis tools to new environments.
>
> > Data split and leakage concern.
>
> The data was randomly split using Python’s train_test_split() function, ensuring no overlap between the train, validation, and test sets. Class distribution is balanced (60% positive, 40% negative). Train and validation sets were not used for fine-tuning, as we focus on evaluating integration without labeled training data. Therefore, evaluation was done only on the test set, avoiding data leakage.
>
> > T2 results are expected, as LLMs struggle with spatial reasoning and heuristics naturally improve their performance.
>
> While heuristics perform well enough on their own (conditioned on tedious threshold search),  **we had little indication prior to this research of how effectively LLMs would use those heuristics.** Our findings show that LLMs do not rely on fixed thresholds but rather dynamically adjust thresholds per instance, a novel insight into how LLMs integrate heuristic information for spatial reasoning tasks. We also note that, as described below, our results with multimodal models indicate that heuristic-based prompting outperforms not only text-based prompting but also prompting with VLM models, further indicating the efficacy of this approach.
>
> **TBC...**

---

> > ### Author Response · Authors · 2025-06-02
> > **Rebuttal by Authors (Continued)**
> >
> > > Add visualizations as input would be a stronger baseline than just textual coordinates.
> >
> > We agree that visual information could potentially help LLMs in spatial integration tasks, as images offer direct cues about orientation and proximity. **To investigate this, we conducted preliminary experiments using vision-language models (VLMs) on the two tasks.**
> >
> > We evaluated one open-source model (Qwen-2.5-VL-7B-Instruct) and the same two API-based models (gpt-4o-mini and gpt-4o, for comparison purpose) in a zero-shot setting, providing only the task description. Each model followed the same instructions as in our heuristic prompting setup and generated binary labels without explanations, using visual inputs like the image in Figure 2. The results below compare three settings: “text” (text-only prompting), “text(features)” (heuristic prompting with features), and “text+img” (text plus images).
> >
> > **Spatial join**
> > | models | text | text+img | text(features) |
> > |--|--:|--:|--:|
> > | qwen-2.5-vl | - | 56.6% | - |
> > | 4o-mini | 40.5% | 62.7% | 93.8% |
> > | 4o | 58.0% | 72.8% | 94.8% |
> > |…|…|…|…|
> >
> > **Spatial union**
> > | models | text | text+img | text(features) |
> > |--|--:|--:|--:|
> > | qwen-2.5-vl | - | 38.3% | - |
> > | 4o-mini | 45.4% | 62.4% | 79.7% |
> > | 4o | 81.7% | 88.9% | 95.5% |
> > |…|…|…|…|
> >
> > While visual input improved results compared to text-only prompting, heuristic prompting with geometric features still outperforms prompting with visual inputs. **This suggests that, although visual context helps, providing geometric features remains highly useful for these tasks.** Overall, we think that adding visual information for spatial tasks is a promising direction and plan to explore more thorough multimodal approaches in future work.

---

> > > ### Comment · Reviewer_ap53 · 2025-06-06
> > >
> > > I appreciate the detailed response, and for taking my feedback into account. To respond to a few main points:
> > > * Data split and leakage concern.
> > > This has not been addressed by your use of `train_test_split()`. There is still possible pseudo-leakage. For example, suppose there are 3 roads {A,B,C} in neighborhood Y. It is possible with your random splitting that {A,B} is in train, {B, C} in test, and {A,C} in dev. This is partial leakage in that the train has seen coordinates for road A and B, and when eval/testing it sees them again, albeit with different paired roads.
> > > * "This limits the generalizability of heuristic-based methods." Am I correct in saying that your best-performing method still uses heuristics, albeit in combination with LLMs? If so, then the manual effort is still being used and LLM's aren't reducing that effort, only boosting performance.
> > > * Thank you for providing some initial experiments with the Qwen 2.5 VLM. This is very interesting and does address my concern that VLMs could solve the task, which as you show here is not necessarily the case. I encourage you to push in this direction for followup work.
> > > * The additional 4 spatial experiments  in your other response are also insightful, but they only add additional perspectives on the geospatial domain, so the full exploration is limited in my view.
> > >
> > > I have raised my score from 3 to 5, but I still think the major changes suggested by the reviewers require another round of revision and reviews.

---

> > > > ### Author Response · Authors · 2025-06-09
> > > >
> > > > > Data split and leakage concern.
> > > >
> > > > Thank you for following up on this concern.
> > > >
> > > > The training and validation sets were not used for fine-tuning in our study, as our methods assess the model's ability to integrate spatial relationships without learning from labeled training examples. We nonetheless divided the data into traditional train/val/test sets to 1) allow for descriptive statistics provided in the appendix and 2) allow for future studies to compare LLMs or other ML models trained on labeled examples against our approach. The evaluation in our paper, however, is conducted exclusively on the test dataset, without learning from the training set. Therefore, even if pseudo-leakage could theoretically occur, it would not practically impact the results or conclusions of this paper.
> > > >
> > > > In the context of future work that might use our dataset, we also note that even if roads A and B appear in the training set (as in your example), the model does not learn their coordinates in isolation. Instead, it learns spatial relationships exclusively between pairs, such as (A,B). The spatial integration task evaluates the model's understanding of these relationships rather than individual object coordinates. Moreover, importantly, the model never encounters road C in the training set, such that spatial relationships derived from new combinations like (A,C) or (B,C) remain unseen during training. For instance (using the simply sketched examples below; "0" represents a node), if the model learns from pair (A,B) that parallel segments indicate a positive spatial relationship, it does not inherently know anything about the spatial relationships involving roads A and C or roads B and C. Relationships such as (A,C) or (B,C) present new spatial configurations to the model during testing.
> > > >
> > > > ```
> > > > A:                       B:                           C:
> > > > O---O                 O---O                            O
> > > >                        \                              /
> > > >                         \                            /
> > > >                          O                          O
> > > >                                                     |
> > > >                                                     |
> > > >                                                     O
> > > > ```
> > > > Thank you again for allowing us to clarify this matter.
> > > >
> > > > > "This limits the generalizability of heuristic-based methods." Am I correct in saying that your best-performing method still uses heuristics, albeit in combination with LLMs? If so, then the manual effort is still being used and LLM's aren't reducing that effort, only boosting performance.
> > > >
> > > > You’re correct to say that the best-performing method still uses heuristics, in combination with LLMs; however, the method saves more manual effort than might be immediately apparent.
> > > >
> > > > Heuristic-based methods involve three main steps: (1) selecting appropriate heuristics, (2) calculating heuristic features, and (3) determining feature thresholds.
> > > >
> > > > The step requiring the most manual effort is Step 3. Identifying optimal thresholds, which requires expensive labeled data collection for each task and involves significant tuning and testing, is time-consuming, requires expertise on the part of the user, and can introduce variability in performance outcomes, especially as the number of heuristic features increases. Using LLMs entirely eliminates the process of threshold selection, as we provide a precomputed heuristic feature to the LLM, but we do not provide any information about what threshold the LLM should use. We rely on LLMs to determine the utility of each heuristic feature based on their understanding of the spatial environment, without explicit human input with respect to appropriate thresholds.
> > > >
> > > > Although Step 1 (selecting which heuristics to compute) may still require some human involvement, this effort is comparatively minimal. Additionally, the computation of heuristic features in Step 2 requires only minimal effort. Thus, our approach leveraging LLMs significantly reduces the manual effort associated with heuristic-based methods, specifically on Step 3, by eliminating the expert-driven threshold determination process, while also improving performance.
> > > >
> > > > All that said, we do agree that further reducing human effort is a worthwhile objective. For example, future work might evaluate providing all possible heuristics to LLMs and allowing the models to decide which ones to use; step 2 might further benefit from frameworks that integrate programming and LLM calls. We see this as a promising direction for future work.

---

> > > > > ### Comment · Reviewer_ap53 · 2025-06-09
> > > > >
> > > > > Thank you for the detailed clarification that "The training and validation sets were not used for fine-tuning in our study, as our methods assess the model's ability to integrate spatial relationships without learning from labeled training examples."
> > > > >
> > > > > That is good to know. Can I clarify if for the few-shot prompts, you used the exact prompts in Table 13, 14, etc., with the 2 given exemplars that are always the same? Since you did not specify in your paper how the few-shot exemplars were selected, I had assumed that you took them from the train set, which leads to the possible leakage concern. However, if you always used these 2 exemplars for all prompts, then that issue should be satisfactorily addressed, provided you remove those from all the train/dev/test splits, and update the text to say these are "static few-shot exemplars". And your illustrative spatial example is reasonable to me.

---

> > > > > > ### Author Response · Authors · 2025-06-10
> > > > > >
> > > > > > We are glad to clarify things out!
> > > > > >
> > > > > > Yes, you are correct. The same two examples (for each task) are used for all prompts. They were removed from all splits, avoiding any leakage. We will update the text accordingly to reflect this.
> > > > > >
> > > > > > Thank you!

---

### Official Review · Reviewer_boFx · 2025-05-10

**Rating:** 7
**Confidence:** 4
**Ethics Flag:** 1

**Summary:**

This paper investigates two map matching tasks where geometrical knowledge is critical and therefore LLMs do poorly.   Good human-generated heuristics exist for these two tasks but thresholds for such heuristics are fragile and tuning them does not scale well.   This paper shows that LLMs can perform extremely well if given some relevant feature values (e.g. separation between two curves).   This gives the LLM a strong sense of direction and relieves it of the task of doing geometrical calculations, but still allows the LLM to do part of the job it does well:   selecting among the given features and dynamically tuning their thresholds for the specific query.

Experiments are run with a range of LLMs and different choices for the prompting approach.

**Questions To Authors:**

Your use of the term "topological" is oddly out of sync with standard mathematics, where your features would all be called "geometrical," contrasting with "topological" features which don't care about things like distance.  I'm guessing it's traditional in this sort of map processing.   If so, it's ok but you should include a very brief comment about what it means in mapping world or even just that mapping world uses it different from math world.

Figure 1 doesn't make much sense until the features are defined in section 3.    I don't see an easy short way to make clear on in section 1 what, say, min-angle means in the figure caption.   So I'd suggest moving the figure near the end of section 3.   Also, the road in that example seems to have zero length.   If it's not a typo, maybe pick a different example where both map elements have visibly different endpoint numbers.

The background colors in Fig 2 are very faint.   Maybe improve the contrast even if you also have to make the road/sidewalk lines a bit wider/darker.

You're modeling features like roads as one-dimensional.   I suspect this is also traditional for the task, e.g. the map information doesn't include the width of each road.   But a couple words would make this clearer to the reader.   This becomes particularly important when you define max_area and we wonder why you have a magic buffer width that is partly compensating for the missing widths.

When you talk about why the matching task is hard (section 3), you could call out changes over time between the two information sources.   Also notice that many roads have a bike path between the road and the sidewalk, which is an even worse problem for blind pedestrians because there is frequently less of a barrier separating it from the sidewalk.

References to Table 8 in section 5:  on the first one, add a couple words saying it's in the appendix.

Line 323 has a typo "hresholds".

**Reasons To Accept:**

Combining an LLM with a symbolic engine or human help is, of course, not new.   This seems like a really good way to divide the task between the LLM, the geometry engine, and the human domain expert.    The empirical results are extremely strong.    The ideas should extend to a much wider range of tasks.   The paper is well written and easy to understand.

**Reasons To Reject:**

They test on only two map-analysis tasks.   I think this is a weak reason to reject because the design seems very likely to generalize and it's more important to get the ideas out where other folks can use them.

---

> ### Author Response · Authors · 2025-06-02
> **Rebuttal by Authors**
>
> We appreciate the reviews’ thoughtful questions. Here are our responses:
>
> > The term "topological" is used in a non-standard way.
>
> Thank you for the clarification and suggestion. Features like distance, angle, and area are better described as geometrical rather than topological in the mathematical sense. We’ll revise accordingly throughout the paper.
>
> > Figure 1 is unclear without feature definitions and has typos
>
> We’ll make the following improvements to the figure: (1) move it to the end of Section 3, where the geometric features are defined, and (2) correct the typo in the road coordinates. To improve clarity, we’ll also add a visual example to make the feature definitions more intuitive.
>
> > The background colors in Fig 2 are very faint.
>
> Thanks for this feedback, we will make the background clearer.
>
> > Clarify that roads are modeled as one-dimensional and explain the use of buffer widths.
>
> Thanks for this suggestion, we will add some text to clarify this. The roads and sidewalks in our dataset are represented as centerlines, which as you observe, makes them one-dimensional. It is standard in spatial data sources like OpenStreetMap.  You are also correct that the 10-meter buffer used in preprocessing helps approximate real-world road widths.
>
> > Highlight temporal changes and bike paths examples as extra challenges for the tasks
>
> We agree that changes over time, like road expansions, can alter widths and buffers, causing heuristics to fail. The presence of bike paths between roads and sidewalks also adds complexity, especially for blind pedestrians. We will incorporate these points in our discussion.
>
> > Mention that Table 8 is in the appendix, and fix the typo "hresholds" on line 323.
>
> We will add the clarification and fix the typo. Thank you.
>
> > Testing on just two tasks is a weak reason to reject.
>
> We agree that the tasks are representative of the spatial domain and likely to generalize. However, as the question of whether two tasks is enough came up in several other reviews, we have  conducted additional experiments on four more spatial analysis tasks involving both spatial reasoning and geometric computation.
>
> **Due to space constraints, we have included these results in the response to Reviewer VhgA. Depending on length considerations, we will include these limited validations either in the discussion of the paper or in an appendix.**

---

> > ### Comment · Reviewer_boFx · 2025-06-07
> > **Reply to author comments**
> >
> > I'm lowering my rating slightly based on some issues that others identified.
> >
> > It looks like there's a lot of issues that are very fixable, esp. details that could be made more clear.   If this is accepted, I strongly urge you to address as many as you can in the available time, since this will make your ideas accessible to a wider audience.

---

### Official Review · Reviewer_VhgA · 2025-05-16

**Rating:** 6
**Confidence:** 2
**Ethics Flag:** 1

**Summary:**

This work investigate LLM's abilities in spatial reasoning tasks, and proposes specific prompting methods and review-and-refine agent workflow to improve.

**Questions To Authors:**

What are the relationship with works about LLMs as world model?

**Reasons To Accept:**

The writing seems to be clear and easy to follow.

**Reasons To Reject:**

1. The scope is narrow to specific spatial reasoning task.

2. The proposed method including heuristic prompts and review-and-refine are all established methods that are widely used in agent-based research. And this work seems to just apply these in spatial reasoning tasks.

---

> ### Author Response · Authors · 2025-06-02
> **Rebuttal by Authors**
>
> We appreciate the reviews’ thoughtful questions. Here are our responses:
>
> > What is the relationship of this study with works about LLMs as world models?
>
> Thank you for this perspective. While our study does not directly engage with the literature on LLMs as world models, there may be conceptual similarity, especially in examining whether LLMs can reason about spatial relationships grounded in real-world structures:
>
> 1. While existing studies on LLMs as world models often emphasize physical dynamics or simulation-based understanding, we focus specifically on spatial reasoning over structured geographic data, such as roads, sidewalks, and polygons.
>
> 2. Rather than simulating general physical environments, we assess whether LLMs possess sufficient latent world knowledge to perform basic spatial integration tasks without explicit training or labeled data. In that sense, we could view this work as an early step toward expanding an LLM’s world model to encompass geospatial contexts and structured environmental data, which are important for urban applications.
>
> > The scope is narrow to specific spatial reasoning tasks.
>
> **Our study is application-driven and aims to provide a fundamental empirical evaluation of LLMs on two core spatial integration tasks that serve as a foundation for many more complex spatial tasks.** Our work is not intended to serve as a comprehensive investigation into LLMs’ spatial reasoning abilities; rather, we know from much prior work in urban spatial analysis that capability in these tasks underlies much of what practitioners hope to accomplish in more challenging scenarios.
>
> We do agree that evaluating a broader range of spatial tasks would yield more general insights into LLMs’ spatial capabilities. Thus, we conducted additional experiments on four spatial analysis tasks that vary in geometry types, computational demands, and reasoning logic. **These new experiments broaden the scope of our evaluation while maintaining coherence within the spatial analysis domain.** We will add the results as supplementary/ablation study.
>
> - Point-to-Segment Distance (P2S): Compute the shortest distance between a point and a line segment.
> - Spatial Containment (SC): Determine whether a point lies inside a polygon (triangle).
> - Spatial Intersection (SI): Check if two line segments intersect.
> - Convex Hull (CH): Identify the minimal set of points forming the convex hull of a point set.
>
> Each task was evaluated on 50 synthetic samples with geometries within a unit square. We use simple settings for preliminary experiments. We evaluate qwen-plus, 4o-mini, and 4o, and report accuracy. For P2S, a tolerance of 1e-3 was used for correctness. Results are below:
>
> | models | P2S | SC | SI | CH |
> |--|--:|--:|--:|--:|
> | qwen-plus | 94% | 82% | 80% | 62% |
> | 4o-mini | 82% | 76% | 64% | 46% |
> | 4o | 94% | 84% | 66% | 64% |
>
> Models performed well on simpler tasks (P2S: 90%, SC: 81%) but struggled with more complex ones (SI: 70%, CH: 57%). A brief qualitative analysis of GPT-4o’s outputs revealed encouraging signs of task understanding and spatial reasoning ability. For instance, it consistently applied projection logic for P2S, ray casting for SC, orientation-based reasoning for SI, and referenced Graham’s scan or Jarvis march in CH. However, computations errors often led to incorrect answers.
>
> **These experiments offer preliminary but valuable insight suggesting that LLMs’ spatial reasoning capabilities extend beyond basic integration tasks.** We believe that our results on spatial join and union, along with these four supplementary tasks, provides meaningful and a broader scope of insight into the fundamental capabilities and limitations of LLMs in applications involving spatial data analysis. They motivate further evaluation to include real-world geometries and more complex task variants in ongoing work.
>
> > The method relies on well-established techniques offering little novelty.
>
> We do not view the primary contribution of our work as a novel methodology. **Our main contribution lies in conducting a comprehensive empirical evaluation of LLMs on spatial integration tasks, an area that, to the best of our knowledge, remains unexplored.** This study serves as an initial step toward understanding how LLMs can be effectively applied to applications involving spatial data.
>
> In addition, we note that our analysis of spatial reasoning and geometric computation differs from prior work in both domains. Existing spatial reasoning studies often center on visual data and tasks like relative positioning or navigation, while geometric reasoning research tends to focus on abstract shapes and textbook math problems (like analyzing triangles and circles). **In contrast, we analyze LLMs’ performance on tasks involving real-world geometries, such as linestrings and polygons, which better reflect practical challenges and offer new opportunities for application-driven research, including in areas like urban accessibility.**

---

> > ### Comment · Reviewer_VhgA · 2025-06-06
> >
> > I appreciate the authors for their rebuttal and for clarifying the application-driven focus of their work. The paper presents a thorough evaluation and offers valuable insights into spatial tasks. However, as I have limited expertise in spatial analysis, I am not in a position to fully assess the contribution. Nonetheless, I have raised my score.

---

### Official Review · Reviewer_xnXe · 2025-05-23

**Rating:** 6
**Confidence:** 3
**Ethics Flag:** 1

**Summary:**

This paper presents a thorough empirical investigation into the use of LLMs for urban spatial data integration. The authors provide valuable insights and propose practical prompting strategies, such as the review-and-refine method. They demonstrate how LLMs can assist in tasks like spatial join and union, offering clear utility for domain experts. However, the methodology primarily builds on existing LLM capabilities and relies on relatively straightforward heuristic augmentations. While the empirical results are compelling, the paper lacks methodological innovation or theoretical advancement in spatial reasoning or model design. Overall, it is a solid and informative contribution to the field, particularly as an early step toward integrating LLMs with spatial data applications.

**Questions To Authors:**

1. Line 225: The model descriptions include strings like “GPT-4o, 2024-08-06 (4o).” Could you clarify whether this notation is meant to indicate the model version?
2. Table 1: Section 4.1 states: “For join, we evaluate three duo heuristics {(p,c), (p,o), (c,o)}” However, in Table 1, it’s unclear whether the duo results correspond to one specific duo heuristic or are averaged across all three.
3. Line 568: Road widths can vary significantly (e.g., residential vs. highways) Is the use of a fixed 10m buffer appropriate across all road types?
4. You mention that LLMs struggle with geometric computations. Given this limitation, have you considered approaches like ViperGPT, which use code generation to compose modular visual and spatial reasoning tools via Python APIs? Such frameworks could potentially offload spatial computations to explicit routines, improving both interpretability and generalization without additional training.

**Reasons To Accept:**

1. The paper addresses an emerging and practically relevant problem — applying LLMs to urban spatial data integration — which is underexplored yet highly impactful in real-world domains.
2. It provides a solid empirical evaluation of LLMs on spatial tasks, demonstrating the utility of heuristic prompting and review-and-refine strategies in improving performance without labeled data.
3. The work shows how domain experts can use LLMs with minimal engineering effort, highlighting the potential for accessible, low-resource spatial data integration tools.
4. The paper is well-written and easy to follow, with clear experimental setups and useful visualizations, making it a good reference for practitioners exploring LLMs in geospatial domains.

**Reasons To Reject:**

1. The paper lacks a dedicated method section, and the methodological descriptions are scattered and limited in depth. This weakens the theoretical grounding of the work and reflects a lack of methodological novelty.
2. All experiments focus on only two relatively simple tasks (spatial join and union), which raises concerns about whether the model truly understands spatial relationships. The approach appears highly dependent on task-specific input formatting, suggesting limited generalization and insufficient task modeling capabilities.
3. Model performance drops significantly (to ~50–60% accuracy) without geometric features and improves drastically (to 90–99%) when such features are provided. However, the paper lacks a detailed analysis of this large performance gap, missing an opportunity to investigate what the model is actually learning.

---

> ### Author Response · Authors · 2025-06-02
> **Rebuttal by Authors**
>
> We appreciate the reviewer’s thoughtful questions. Here are our responses:
>
> > Is the notation “GPT-4o, 2024-08-06 (4o)” used to indicate the model version?
>
> Yes. “GPT-4o, 2024-08-06” refers to the specific model version gpt-4o-2024-08-06. The shorthand “(4o)” is used for brevity in presenting results, as with “(4o-mini)” and “(llama3).”
>
> > Does Table 1 show results for a specific duo heuristic or an average of all three?
>
> It is the average over all three duo combinations. We provide Individual results below, showing consistent performance and close alignment with the overall average.
>
> | duo | worst | best | avg (std.) |
> |--|--:|--:|--:|
> | (p,c) | 0.837 | 0.951 | 0.915 (0.04) |
> | (p,o) | 0.735 | 0.909 | 0.909 (0.08) |
> | (p,c) | 0.758 | 0.972 | 0.901 (0.06) |
>
> > Is using a fixed 10m buffer suitable for all road types given the variation in road widths?
>
> The 10m buffer reflects the origins of our dataset in accessibility research, which is typically concerned only with roads that have sidewalks. Thus, while there is substantial variation in real-world road widths, our dataset includes only road types likely to have adjacent sidewalks – {primary, secondary, tertiary, residential, living_street}, and excludes highways. The selection of the 10-meter buffer is based on typical road widths in Seattle and Bellevue, and applied across all types.
>
> > Have you considered approaches like ViperGPT, which use code generation to compose modular visual and spatial reasoning tools via Python APIs?
>
> We haven’t tested ViperGPT or similar approaches, though we agree that these frameworks could be beneficial for addressing some of the computational limitations our work identifies. The computation of the geometric features used in this paper requires very light human programming, rendering feasible a purely natural language approach with geometric features supplied in the prompt. However, where we want to perform spatial tasks that require deriving complex features, or combining information from several data sources, we expect that the benefits of a framework that integrates programming and natural language instructions will become more clear. We see this as a promising direction for future work.
>
> > The paper lacks a dedicated method section, which weakens the theoretical grounding of the work and reflects a lack of methodological novelty.
>
> 1. We combined methods and experimental setup to conserve space; we will distinguish them in the revision and emphasize the theoretical grounding related to limitations of spatial and geometric reasoning as distinct from other tasks.
>
> 2. **Our main contribution lies in conducting a comprehensive empirical evaluation of LLMs on spatial integration tasks, an area that, to the best of our knowledge, remains largely underexplored.** This study serves as an initial step toward understanding how LLMs can be effectively applied to applications involving spatial data.  We believe that our findings provide valuable insights and establish a foundation for future research that may introduce more methodologically innovative solutions.
>
> > All experiments focus on only two relatively simple tasks (spatial join and union), which raises concerns about whether the model truly understands spatial relationships.
>
> 1. While the two tasks may appear simple, **they are foundational to more complex spatial analysis, and focusing on just these ubiquitous, fundamental tasks provides us with a highly interpretable view of  LLMs’ basic capabilities in handling spatial tasks.** In fact, our results show that even these foundational tasks pose a notable challenge for LLMs, in that they reveal the models’ computational limitations.
>
> 2. Nonetheless, we agree that evaluating a wider range of tasks can provide deeper and more general insight into LLMs’ capabilities. In response, we conducted additional experiments on four more spatial analysis tasks involving both spatial reasoning and geometric computation. **Due to space constraints, we refer the reviewer to our response to Reviewer VhgA's second question for details on these supplementary results.**
>
> > The paper lacks a detailed analysis of the large performance gap (hints v.s. features)
>
> We address this in Section 5.3 with a detailed qualitative analysis of how LLMs approach both tasks with and without geometric features.
>
> 1. Without features, LLMs attempt to infer spatial relationships by assessing proximity and alignment between pairs. While this shows some spatial understanding, they often fail due to poor execution of geometric calculation, including incorrect logic and flawed calculations.
>
> 2. In contrast, providing heuristic features fundamentally shifts the nature of the task. Rather than performing geometric calculation, models simply evaluate the provided feature values. This reduces the task to a rule-checking problem, resulting in  better accuracy.

---

> > ### Comment · Reviewer_xnXe · 2025-06-07
> >
> > The authors have provided a proactive and detailed rebuttal that addresses most concerns regarding the experimental setup and explanatory clarity. While the methodological novelty remains limited, the work offers clear practical value in applying LLMs to geospatial data integration and establishes a solid foundation for future research. The clarifications and additional analyses have increased confidence in the submission. The original score of 6 is therefore maintained, with a stronger inclination to support acceptance — particularly if the final version improves structural organization and incorporates the additional experimental results.

---

### Decision · Program_Chairs · 2025-07-08

**Decision:**

Accept

**Comment:**

The paper investigate the use of LLMs to integrate urban spatial data from multiple datasets, focusing on two task: 1) spatial join - associating spatial elements [e.g sidewalk next to road(, and 2) spatial union - identifying when two elements partially / fully represent the same entity in the real-world.   Datasets for the two tasks are obtained from a public data source, and experiments investigates the performance of different methods and variations, comparing heuristic based methods with different LLM-prompting methods.

**Strengths**
Reviewers noted that the following:
- The work addresses an interesting problem of spatial data integration, which hasn’t been studied before [xnXe]
- The division of work between LLM, geometry tool, and human is good [boFx]
- The evaluation of LLMs and comparison to heuristics is fairly thorough [xnXe,ap53] with interesting analysis [ap53] and strong empirical performance [boFx]
- The paper is mostly well written and clear [xnXe, VhgA, boFx, ap53]

**Weaknesses**
The main concern from reviewers that the work only studies two fairly simple tasks [xnXe, VhgA].  This raises some concerns that generalizability of the method is not demonstrated.

Other concerns from reviewers includes:
- Some improvements to the manuscript and presentation is needed [xnXe, boFx, ap53] as there are currently unclear details [xnXe], inaccurate terminology [boFx], and some other issues with the exposition [boFx, ap53].
- Lack of visual input for LLMs [ap53]

Most of these weaknesses were addressed by the authors.  During the rebuttal, the authors included results on several additional tasks (position-to-segment distance, spatial containment, spatial intersection, convex hull) in response to R-VhgA, and use of visual input (for R-ap53).  Most reviewers are positive on this work, with R-VhgA and R-ap53 increasing their scores after the rebuttal.  R-ap53 indicated that they felt the work would benefit from another round of revisions and review, and gave the work a 5 (marginally below acceptance).  Other reviewers are in favor of acceptance.

**Recommendation**
The AC believe the work is sufficiently solid and that updates to improve the paper and incorporate additional experimental results can be done in a minor revision.  The AC thus recommend acceptance.  The authors should make sure the improve the exposition and figures following reviewer feedback, and to incorporate the additional experimental results in the camera ready.